# ADiff4TPP: Asynchronous Diffusion Models for Temporal Point Processes

**Amartya Mukherjee**[*][†]                                           *a29mukhe@uwaterloo.ca*
*Department of Applied Mathematics, University of Waterloo*
*RBC Borealis*

**Ruizhi Deng**                                                      *ruizhi.deng@borealisai.com*
*RBC Borealis*

**He Zhao**                                                          *he.zhao@borealisai.com*
*RBC Borealis*

**Yuzhen Mao**[†]                                                    *yuzhenm@sfu.ca*
*School of Computing Science, Simon Fraser University*
*RBC Borealis*

**Leonid Sigal**                                                     *lsigal@cs.ubc.ca*
*Department of Computer Science, University of British Columbia*
*RBC Borealis*

**Frederick Tung**                                                   *frederick.tung@borealisai.com*
*RBC Borealis*

**Reviewed on OpenReview:** *https://openreview.net/forum?id=bwnZW4wXh4*

## Abstract

This work introduces a diffusion model-based approach to modelling temporal point processes via an asynchronous noise schedule. Existing methods typically rely on parametric conditional intensity functions or autoregressive next-event prediction, which can limit distributional expressivity and make long-horizon forecasting computationally expensive. We address this limitation by using diffusion models to learn the joint distribution of event sequences in latent space without imposing restrictive parametric assumptions. At each step of the diffusion process, the noise schedule injects noise of varying scales into different parts of the data. With a careful design of the noise schedules, earlier events are generated faster than later ones, thus providing stronger conditioning for forecasting the more distant future. We derive an objective to effectively train these models for a general family of noise schedules based on conditional flow matching. Our method models the joint distribution of the latent representations of events in a sequence and achieves state-of-the-art results in predicting both the next inter-event time and event type on benchmark datasets. Additionally, it flexibly accommodates varying lengths of observation and prediction windows in different forecasting settings by adjusting the starting and ending points of the generation process. Finally, our method shows strong performance in long horizon prediction tasks, outperforming existing baseline methods. The code is available at https://github.com/BorealisAI/adiff4tpp.

---

[*]Corresponding author. [†]Work done during internship at RBC Borealis.

# 1 Introduction

Event sequences are prevalent in many domains, including commerce, science, and healthcare, with event data generated by human activities and natural phenomena such as online purchases (Ni, 2018; Wong, 2014; Xue et al., 2022), social media (Zhou et al., 2013; Leskovec & Krevl, 2014), disasters (Ogata, 1988; Venzke, 2024), and disease outbreaks (Lorch et al., 2018; Rizoiu et al., 2018). Temporal point processes (TPPs) have been a powerful tool to model the distribution of event occurrence over time (Mei & Eisner, 2017; Zuo et al., 2020; Shchur et al., 2020).

Diffusion models (DMs) have emerged as a powerful framework in generative modelling, achieving remarkable success in domains such as image synthesis (Rombach et al., 2022; Sauer et al., 2024; Peebles & Xie, 2023; Zhang et al., 2023) and video generation (Ho et al., 2022b;a; Ma et al., 2024; Blattmann et al., 2023; Khachatryan et al., 2023). Despite these advancements, their application to modelling TPPs remains limited because the current diffusion paradigm faces several challenges when applied to event sequence data as follows.

**Data heterogeneity.** TPPs often consist of mixed data types, combining continuous attributes (e.g., timestamps or durations) with discrete variables (e.g., event categories or marks). Though DMs are a good fit for continuous data, recent work on applying them to discrete data often involve non-trivial efforts (Austin et al., 2021; Inoue et al., 2023).

**Synchronous noise.** Existing DMs typically assume the same level of noisiness in different parts of data at each step of the diffusion or generation process. However, in TPPs, subsequent events could be triggered by previous ones. Thus, reducing the uncertainty in earlier events could lead to better predictions of later ones. This can be achieved by denoising earlier events faster than the later ones, or equivalently diffusing later events faster than the earlier ones. Yet, such asynchronous diffusion strategies can not be realized given the synchronous noise assumption.

**Fixed data dimension.** The length of sequential data is inherently variable. For example, both the context length and prediction horizon in TPP sequences can vary in different problem settings (e.g.,

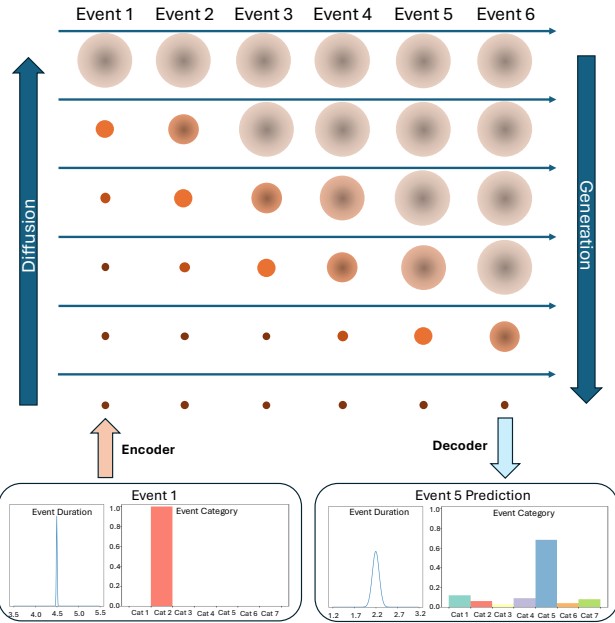

Figure 1: **Asynchronous Noise Schedule.** The size and blurriness of each blob indicates the scale of noise at each event. In the diffusion process (bottom to top), the latent representations of later events are replaced by Gaussian noise faster than earlier ones. In the generation process (top to bottom), early events are denoised first, providing context for the generation of future events.

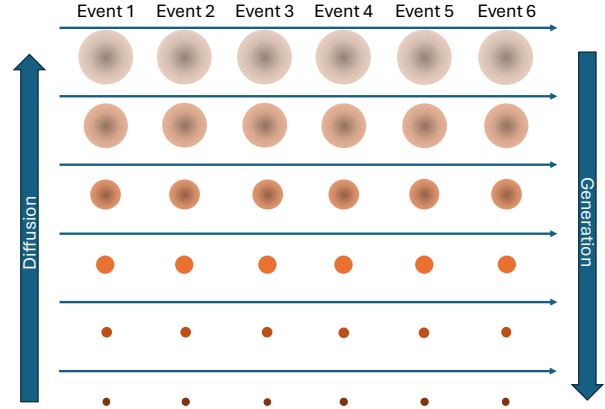

Figure 2: **Synchronous Noise Schedule.** This figure visualizes the synchronous diffusion process when being applied to model TPPs. At each intermediate step in the diffusion or generation process, the model assumes the same scale of noise for each event.

prediction horizon can be one or many for next event and long horizon predictions respectively). DMs typically assume a fixed data dimension, which is unsatisfactory for the demands of TPPs.

We present a design of DMs for TPPs that directly addresses the challenges in three ways. (i) Inspired by latent DMs (Zhang et al., 2024; Rombach et al., 2022), we first learn continuous latent representations that

can reconstruct heterogeneous event variables faithfully using a variational autoencoder ($\beta$-VAE, Higgins et al. (2017)). Then, DMs are applied to model the joint distribution of event representations in this latent space, from which timestamps and event types are decoded to produce the final event sequence. (ii) We adopt DMs with asynchronous noise schedules which diffuse and generate events in a sequence with different speeds (see Figure 1) and derive the training objectives for a general family of such schedules based on flow matching (Lipman et al., 2023). Our method enables faster generation of earlier events in the sequence to provide stronger guidance for the generation of later ones. It can be seen as an alternative to autoregressive generation or whole sequence diffusion with synchronous noise schedules (see Figure 2). (iii) Finally, the asynchronous noise schedules enable flexible observation and prediction windows. During inference, latent variables corresponding to observed events remain fixed while those in the prediction window are initialized as Gaussian noise; by controlling the start and end flow times for different event indices, the model performs conditional generation of future events given past events without retraining. Our approach demonstrates performance superior to existing methods on both next event predictions and long horizon predictions using TPP benchmark datasets.

## 2 Preliminaries

Our approach uses asynchronous DMs to model temporal point processes in a latent space. We use this section to introduce the problem of temporal point process modelling. It also covers the basics of flow matching (Lipman et al., 2023) upon which our derivation of DM training objectives is based.

### 2.1 Temporal Point Processes

Temporal point process (TPP) is a random process whose realization describes a sequence of discrete event occurrences. A typical sequence of $N$ events sampled from a TPP can be characterized as $\mathbf{z} = \{\mathbf{z}^{(1)}, \mathbf{z}^{(2)}, ... \mathbf{z}^{(N)}\}$ with each event $\mathbf{z}^{(i)} = (\tau^{(i)}, k^{(i)})$ where $i$ is an index indicating the chronological order of events, $\tau^{(i)}$ is the duration between the $i$th event and its predecessor, and $k^{(i)}$ denotes the event semantic category (or mark).

Traditionally, TPP models estimate intensity functions for predicting the next event occurrence, and are optimized by maximizing the log-likelihood of event sequences (Mei & Eisner, 2017). Recent works directly approach the problem by learning the conditional distribution of the next event's time and mark (Shchur et al., 2020). TPP models are commonly evaluated on two tasks:

- *Next event prediction.* Given $\{\mathbf{z}^{(1)}, ..., \mathbf{z}^{(i-1)}\}$ from the test set, a TPP model predicts the immediate next event $\mathbf{z}^{(i)} = (\tau^{(i)}, k^{(i)})$ for $1 < i \leqslant N$.

- *Long horizon prediction.* Given the preceding $m$ events $\{\mathbf{z}^{(0)}, ..., \mathbf{z}^{(m)}\}$, a model predicts the next $h$ events $\{\mathbf{z}^{(m+1)}, ..., \mathbf{z}^{(m+h)}\}$.

Our approach can be seen as part of the broader intensity-free family of TPP models, which aim to capture the underlying event dynamics without explicitly parameterizing an intensity function (Shchur et al., 2020). We grounded our work on the existing literature on TPPs and refer to Section 4.4 for a discussion on the distributional characteristics of our proposed method.

### 2.2 Flow-Based Diffusion Models

Flow-based diffusion models (Lipman et al., 2023; Liu et al., 2023) are a class of DMs with close ties to neural ordinary differential equations (ODEs) (Chen et al., 2018). Compared to discrete time or stochastic differential equation (SDE)-based formulation of DMs (Ho et al., 2020; Song et al., 2021), they enjoy the benefits of simple forward process, simulation-free training, and efficient sampling using ODE solvers. This family of methods form the backbone of notable DMs like Stable Diffusion 3 (Esser et al., 2024).

It can transform simple distributions (e.g., Gaussian noise) to complex ones (e.g., real world data) through solving the ODE

$$\dot{\mathbf{x}}_s = v_\theta(\mathbf{x}_s, s), \tag{1}$$

from $s = 1$ to $s = 0$, where $v_\theta(\mathbf{x}_s, s)$ is a learnable vector field that generates the probability paths interpolating between data $\mathbf{x}_0$ and Gaussian noise $\mathbf{x}_1$, Remarkably, Flow Matching (FM) shows that such a vector field can be constructed by marginalizing over conditional vector fields that are tractable and generate the conditional probability paths interpolating between data and noise with one end fixed (Lipman et al., 2023; Esser et al., 2024). It introduces a straightforward objective to learn the parameters of $v_\theta(\mathbf{x}_s, s)$ by regressing the conditional vector fields, termed Conditional Flow Matching (CFM). Given an interpolation between data and noise,

$$\mathbf{x}_s(\mathbf{x}_0, \boldsymbol{\epsilon}) = \alpha(s)\, \mathbf{x}_0 + \gamma(s)\, \boldsymbol{\epsilon}, \tag{2}$$

where $\alpha(s) : [0, 1] \to [0, 1]$ and $\gamma(s) : [0, 1] \to [0, 1]$ are monotonic and differentiable functions satisfying $\alpha(0) = 1$, $\alpha(1) = 0$, $\gamma(1) = 1$, and $\gamma(0) = 0$. CFM defines a conditional flow, $\psi_s(\cdot|\boldsymbol{\epsilon}) := \mathbf{x}_s(\mathbf{x}_0, \boldsymbol{\epsilon})$, that models the evolution of the underlying distribution $\mathbf{x}_s$, and derives the conditional vector field,

$$u_s(\mathbf{x}_s|\boldsymbol{\epsilon}) = \psi'(\psi^{-1}(\mathbf{x}_s|\boldsymbol{\epsilon})|\boldsymbol{\epsilon}),$$

which generates probability paths from data $\mathbf{x}_0$ to a given sample of $\boldsymbol{\epsilon}$. Then, $v_\theta(\mathbf{x}_s, s)$ can be trained with the objective

$$\mathbb{E}_{s \sim \mathcal{U}(0,1], \boldsymbol{\epsilon}, \mathbf{x}_0} \left[ ||v_\theta(\mathbf{x}_s(\mathbf{x}_0, \boldsymbol{\epsilon}), s) - u_s(\mathbf{x}_s(\mathbf{x}_0, \boldsymbol{\epsilon})|\boldsymbol{\epsilon})||^2 \right]. \tag{3}$$

Notably, in the case of $\alpha(s) = 1 - s, \gamma(s) = s$, FM Equation 2 is equivalent to rectified flow (Liu et al., 2023).

**Discussion.** In the original FM work (Lipman et al., 2023), $\alpha(s)$ and $\gamma(s)$ are scalar-valued functions. In Section 4, we derive noise schedules with theoretical guarantees that can diffuse data in asynchronous paces by extending the scalar coefficients $\alpha(s)$ and $\gamma(s)$ to matrices.

## 3  Asynchronous Diffusion for Event Sequences

In this section, we provide details on our methods to perform generation and prediction with asynchronous noise schedules in the presence of mixed data types. We approach this challenge by training a VAE that maps events to a latent space, where we perform diffusion. We introduce diffusion models with a piecewise linear asynchronous noise schedule and provide the training objectives for the DMs. A modified diffusion transformer (DiT) takes an asynchronous noise schedule $A(s)$ as an input instead of the timestep $s$ that is usually used. The entire pipeline of our approach is presented in Figure 1. Finally, we show how we adapt the generative ODEs in asynchronous DMs for different future event prediction tasks. Our pipeline and code for ADiff4TPP is adapted from the implementation of rectified flow by Lee et al. (2024). This bridges the work of Liu et al. (2023) and EDM (Karras et al., 2022) to improve the quality of generated images with fewer function evaluations.

### 3.1  Event Latent Space Representations

Modelling TPPs with both continuous inter-event times and categorical marks poses a fundamental challenge for DMs. DMs are most naturally defined on continuous spaces with smooth geometry, whereas event sequences combine heterogeneous components with a vastly different structure. We approach this challenge by learning a continuous latent representation via a VAE, which embeds events into a continuous space where smooth interpolation and asynchronous noise schedules are well defined. Specifically, we train a $\beta$-VAE with an encoder $\mathbf{E}_\phi(\cdot)$ mapping each $\mathbf{z}^{(i)} = (\tau^{(i)}, k^{(i)})$ to a Gaussian distribution $\mathbf{X}^{(i)} = \mathbf{E}_\phi(\mathbf{z}^{(i)})$ defined in the latent space with mean value denoted by $\mathbf{x}^{(i)}$. Its variance is used only in the training of the VAE, within the KL divergence term to regularize the latent distribution. A decoder $\mathbf{D}_\phi(\cdot)$ is trained to decode $\mathbf{x}^{(i)}$ into a reconstructed event $\tilde{\mathbf{z}}^{(i)} := (\tilde{\tau}^{(i)}, \tilde{k}^{(i)}) = \mathbf{D}_\phi(\mathbf{x}^{(i)})$. The whole $\beta$-VAE is trained with an objective as

$$\begin{aligned} \mathcal{L}_{VAE}(\mathbf{z}^{(i)}) &= \mathcal{L}_{recon}(\mathbf{z}^{(i)}, \tilde{\mathbf{z}}^{(i)}) + \beta \mathcal{L}_{KL}, \\ &= (\tau^{(i)} - \tilde{\tau}^{(i)})^2 + \text{CrossEntropy}(k^{(i)}, \tilde{k}^{(i)}) + \beta \mathcal{L}_{KL}, \end{aligned} \tag{4}$$

where $\mathcal{L}_{recon}(\cdot, \cdot)$ consists of a mean square error loss for reconstructing the inter-event time $\tau^{(i)}$, and a cross-entropy loss for reconstructing the event mark $k^{(i)}$. $\beta$ is a hyper-parameter tuning the strength of

regularization by KL divergence between $\mathbf{X}^{(i)}$ and a standard Gaussian distribution. In practice, $\beta$ could vary in a range $[\beta_{min}, \beta_{max}]$ during training to achieve a better balance between a compact latent space and encoding faithfulness (Zhang et al., 2024).

The encoder and decoder weights are frozen after training and each event $\mathbf{z}^{(i)}$ in the training data is encoded into its latent representation $\mathbf{x}^{(i)}$. Then, DMs are trained to model the joint distribution of $\mathbf{x} = \{\mathbf{x}^{(1)}, \mathbf{x}^{(2)} \ldots, \mathbf{x}^{(n)}\} \in \mathbb{R}^{n \times d}$, where $n$ is the number of events in the sequence and $d$ is the latent vector size. When $n < N$, we pad zeros to $\mathbf{x}$, where $N$ is a hyperparameter of the maximum length, as $\mathbb{R}^{N \times d}$.

We intentionally adopt a design that defers the modelling of temporal dependencies to the DM instead of the VAE. This separation of concerns offers several benefits. It simplifies the training of the VAE - Appendix E shows that events in held-out test sets could be reconstructed with negligible error from their latent representations. Deferring the temporal dependencies to the DiT also allows better control and flexibility over the observation and prediction window lengths in the sequential generation of events.

## 3.2 Asynchronous Diffusion Models

Given a sequence of encoded latent representations, we define the matrix-valued interpolation between data and noise as

$$\mathbf{x}_s(\mathbf{x}_0, \boldsymbol{\epsilon}) = A(s)\,\mathbf{x}_0 + (I - A(s))\,\boldsymbol{\epsilon}, \tag{5}$$

where $\mathbf{x}_0 := \mathbf{x} \in \mathbb{R}^{N \times d}$ is an event sequence in latent space and $\boldsymbol{\epsilon}$ is a Gaussian noise of the same dimension. $N$ is assumed to be the maximum possible length of event sequences and $A(s) \in \mathbb{R}^{N \times N}$ is a matrix-valued function controlling the diffusion speeds for different parts of the data, i.e., the noise schedule. We make the following choice of a diagonal asynchronous matrix $A(s)$:

$$[A(s)]_{ii} = \text{clip}\left(\frac{s_{end}^{(i)} - s}{s_{end}^{(i)} - s_{start}^{(i)}}, \min = 0, \max = 1\right), \quad \text{where} \quad s_{start}^{(i)} = \frac{N - i}{2N - 1}, \quad s_{end}^{(i)} = \frac{2N - i}{2N - 1}, \tag{6}$$

and $[A(s)]_{ij} \equiv 0$ for $i \neq j$. Such choice of noise schedule indicates that when $s$ is between $s_{start}^{(i)}$ and $s_{end}^{(i)}$, the latent representation of the $i_{th}$ event is being diffused from data to Gaussian noise.

An example of the asynchronous noise schedule when diffusing a sequence with $N = 6$ can be seen in Figure 3. Given a matrix-valued noise schedule $A(s)$, we define our generative process using the following ODE

$$\dot{\mathbf{x}}_s = A'(s)v_\theta(\mathbf{x}_s, A(s)). \tag{7}$$

During training, we sample $\mathbf{x}_s$ at an arbitrary time $s \in [0, 1]$ based on Equation 5 and minimize a CFM objective (Equation 12). We discuss the sufficient conditions for a general family of $A(s)$ which not only ensures we can perform asynchronous generation effectively but also complies with the desirable properties of existing synchronous diffusion models in Section 4 and derive the above training objective. In addition, different types of noise schedules are studied and compared empirically in the ablation study (Section 5.3) to justify our design of the asynchronous noise schedule.

Figure 3: Asynchronous noise schedule for an event sequence of length 6. The noise schedule shows that Event 6 (the latest event in the sequence) is the first to be completely diffused (at flow time $s = \frac{6}{11}$). Event 1 (the earliest event) is the last to start diffusing (at $s = \frac{5}{11}$) and be completely diffused (at $s = 1$). Thus, in reverse-flow time (generation), Event 1 is the first to be restored, and Event 6 is the last to be restored.

## 3.3 Diffusion Transformer (DiT)

Following Peebles & Xie (2023), we implement $v_\theta(\cdot, A(s))$ using a DiT architecture with minor modification: Swapping the patchify and patch embed-

ding components with the latent event representations from $\mathbf{E}_\phi(\cdot)$. Upon initializing the model, we provide a hyperparameter $N$ that decides the maximum length of the sequence of events that the model can generate. The modified embedding function processes $A(s) \in \mathbb{R}^{N \times N}$ by broadcasting its elements with sinusoidal frequencies to generate embeddings that encode the matrix's temporal and structural dynamics. During training or generation of events with sequence length $n < N$, we apply a mask to the multihead self-attention. The choice of masked attention, together with the asynchronous noise schedule, reflects the sequential order of data and also allows the model to flexibly handle variable prediction window length.

### 3.4 Future Event Prediction as Conditional Generation

Event forecasting can be characterized by an observation window $O = \{1, ..., n\}$ that precedes a prediction window $P = \{n+1, ..., n+h\}$, where $h$, satisfying $n + h \leq N$, represents the number of events we want to predict. The goal of the model is to predict events in the prediction window conditioned on events in the observation window. Let $\boldsymbol{\epsilon} = \{\boldsymbol{\epsilon}^{(i)}\}_{i=1}^N$ be an initial Gaussian noise, and let $\mathbf{y} = \{\mathbf{y}^{(i)}\}_{i \in O}$ be a latent space representation of the event sequence in an observation window.

ADiff4TPP introduces a simple and intuitive method for event forecasting by leveraging the asynchronous flow matching objective, which frames event forecasting as a generative process for future events. This objective enables the derivation of a simple ODE to reconstruct preceding latent events. The vector field $\mathbf{f}(\mathbf{x}_s, s)$ is defined element-wise as:

$$f_i(\mathbf{x}_s, s) = \begin{cases} [A'(s)]_{ii}(\mathbf{y}^{(i)} - \boldsymbol{\epsilon}^{(i)}) & i \in O \\ [A'(s)]_{ii}[v_\theta(\mathbf{x}_s, A(s))]_i & i \in P. \end{cases} \tag{8}$$

Since $[A'(s)]_{ii}(\mathbf{y}^{(i)} - \boldsymbol{\epsilon}^{(i)})$ remains constant along $[s_{start}^{(i)}, s_{end}^{(i)}]$, ODE solvers such as Euler's method or Runge-Kutta (RK4) can solve the equation within this domain with zero numerical error. Theoretical insights justifying this vector field and connections to the TPP literature are provided in Appendix 4.4. This ODE is solved from $s = s_{end}$ to $s = s_{start}$, where $s_{end} = \max\{s_{end}^{(i)} : i \in P\}$, $s_{start} = \min\{s_{start}^{(i)} : i \in P\}$ with initial condition $\mathbf{x}_{s_{end}} = A(s_{end})\mathbf{x}^* + (1 - A(s_{end}))\boldsymbol{\epsilon}$, where $\mathbf{x}^* = [\mathbf{y}^{(1)}, ..., \mathbf{y}^{(n)}, \boldsymbol{\epsilon}^{(n+1)}, ..., \boldsymbol{\epsilon}^{(N)}]$. This ensures that all the events in $P$ are initialized to Gaussian noise.

The asynchronous nature of ADiff4TPP allows the ODE to be solved over the shorter timespan $[s_{start}, s_{end}]$ instead of the full interval $[0, 1]$, improving computational efficiency. The indices of the event sequence that proceed the prediction window are masked out. Furthermore, both short and long horizon forecasting is performed by solving the same ODE. This eliminates the need for repetitive next event predictions, which are typically required in existing TPP methods. By avoiding the sequential appending of predicted events to the tensor of preceding events, ADiff4TPP significantly reduces evaluation time, making it a more efficient and scalable solution for TPP forecasting tasks.

## 4 Noise Schedules and Training Objectives

Our proposed approach leverages asynchronous noise schedules to enable more flexible and efficient conditional generation, wherein the model generates future events based on partial observations of preceding data. While generative models with asynchronous noise schedules have been explored in prior works such as Lee et al. (2023) and Chen et al. (2024), our formulation introduces greater flexibility by allowing for customizable noise schedules. To support this, we establish a set of sufficient conditions that the noise schedule must satisfy to ensure the model's ability to perform asynchronous generation effectively. Our contributions in this section are summarized as follows:

1. We extend FM to asynchronous noise schedules and derive conditions on the noise schedule for the flow to remain valid.

2. We relax the conditions on invertibility and differentiability of the flow and show that they still preserve the essential properties of FM.

3. We derive objective functions for the more flexible FM method and establish their mathematical equivalence.

### 4.1 Conditions on the Noise Schedule

Extending the approach of Lee et al. (2023), we introduce a positive semi-definite matrix-valued time-varying coefficient $A(s) \in H^1([0,1]; \mathbb{R}^{n \times n})$ (i.e. every scalar element of $A(s)$ is in the Sobolev space $H^1[0,1]$), satisfying $A(0) = I$ and $A(1) = 0$, to parameterize the interpolation in the FM formulation as shown in Equation 5.

The extension of a single scalar $\alpha(s)$ to $A(s)$ allows us to apply more fine-grained control on the speed of interpolation between data and noise to different parts of data. To derive an objective function, we extend the continuous FM framework by first defining a flow $\psi_s(\cdot|\boldsymbol{\epsilon}) := \mathbf{x}_s(\mathbf{x}_0, \boldsymbol{\epsilon})$. Our notation is based on the work of Esser et al. (2024), where the flow is conditioned on the noise $\boldsymbol{\epsilon}$. We first consider some sufficient conditions for $\alpha(s)$ to satisfy for FM, like boundary conditions, non-negativity, monotonicity, and continuity. Similarly, we introduce the following conditions for $A(s)$.

**Assumption 4.1.** The noise schedule $A(s) \in H^1([0,1]; \mathbb{R}^{n \times n})$ satisfies the following conditions:

1. $A(s)$ satisfies the boundary conditions: $A(0) = I, A(1) = 0$.

2. $A(s)$ is positive semi-definite for all $s \in [0,1]$

3. $A(s)$ is monotone non-increasing in the Löwner order (Löwner, 1934). This means $A(s) - A(s')$ is negative semi-definite if $s \geq s'$, also denoted as $A(s) \preceq A(s')$.

4. $A(s)$ is continuous for all $s \in [0,1]$.

This set of conditions ensure our asynchronous DM aligns with the desirable properties of existing DMs. Please refer to Appendix A.1 for a more detailed discussion on physical interpretability. Subsequently, we introduce a lemma that verifies the validity of the flow provided the conditions.

**Lemma 4.2** (Informal). *The flow $\psi(\mathbf{x}_s|\boldsymbol{\epsilon})$ in Equation 5 governed by a matrix-valued coefficient $A(s) \in H^1([0,1]; \mathbb{R}^{n \times n})$ remains valid as long as $A(s)$ satisfies condition (4) from Assumption 4.1.*

The lemma shows that the FM is well-posed even for asynchronous noise schedules, as the transformation $A(s)$ preserves the essential properties of FM. We refer the reader to Appendix A.4 for a proof of the formal lemma.

### 4.2 Invertibility and Differentiability of the Noise Schedule

Conditional FM (Lipman et al., 2023) characterizes diffusion processes by defining a flow $\psi_s(\cdot|\boldsymbol{\epsilon})$ and the corresponding vector field. This framework operates on the assumption that the interpolation is governed by a known function of $s$. A critical assumption in the FM framework is that the flow is invertible. However, this assumption is not always satisfied in the asynchronous formulation, presenting a significant challenge when extending FM to the asynchronous setting. To address this, we introduce theoretical results that relax this assumption, ensuring the framework remains applicable in scenarios where invertibility does not hold. To characterize the flow $\psi_s(\cdot|\boldsymbol{\epsilon})$ as an ODE, we consult the conditional probability path formula (Lipman et al., 2023, Equation 13), $u_s(\mathbf{x}_s|\boldsymbol{\epsilon}) = \psi'_s(\psi_s^{-1}(\mathbf{x}_s|\boldsymbol{\epsilon})|\boldsymbol{\epsilon})$. We compute $\psi'_s(\cdot|\boldsymbol{\epsilon})$ and define $\psi_s^{-1}(\cdot|\boldsymbol{\epsilon})$ as follows:

$$
\begin{aligned}
\psi'_s(\mathbf{x}|\boldsymbol{\epsilon}) &= A'(s)\mathbf{x} - A'(s)\boldsymbol{\epsilon}, \\
\psi_s^{-1}(\mathbf{x}_s|\boldsymbol{\epsilon}) &:= A(s)^\dagger(\mathbf{x}_s - \boldsymbol{\epsilon}) + \boldsymbol{\epsilon},
\end{aligned}
\tag{9}
$$

where $A(s)^\dagger$ is the Moore-Penrose Pseudo-Inverse of $A(s)$ (Moore, 1920). We refer to Lemma A.3 for a discussion on the validity of $\psi_s^{-1}(\cdot|\boldsymbol{\epsilon})$, notably that it recovers partially diffused components of $\mathbf{x}_s$.

*Remark* 4.3. $A(s)$ may not be differentiable at finitely many points in $[0,1]$. Throughout this paper, we interpret $A'(s)$ as the *weak derivative* of $A(s)$, which is guaranteed to exist if $A(s)$ is continuous and piecewise-differentiable.

### 4.3 Equivalence of Objective Functions

Plugging the derivative and inverse terms into the FM formulation gives us the desired conditional vector field:

$$u_s(\cdot|\boldsymbol{\epsilon}) = A'(s)A(s)^\dagger[\mathbf{x}_s - \boldsymbol{\epsilon}] \tag{10}$$

$$\equiv A'(s)[\mathbf{x}_0 - \boldsymbol{\epsilon}]. \tag{11}$$

Because $A(s)^\dagger$ has multiple asymptotes in $s \in [0,1]$, Equation 10 is ill-posed, so we prefer to solve Equation 11. We refer to Proposition A.5 for a discussion on the equivalence of the vector fields. For verification, we also explore a family of invertible noise schedules that satisfy conditions (2)–(4) in Assumption 4.1, and show that they yield the same marginal vector field as we approach the "non-invertible limit" in Appendix B. Since the asynchronous noise schedule $A(s)$ is decided prior to training, we can simplify the training of our flow model $v_\theta(\mathbf{x}_s, A(s))$ to predict $\mathbf{x}_0 - \boldsymbol{\epsilon}$. This model is trained by minimizing the CFM objective:

$$\mathcal{L}_{CFM}(\theta) = \mathbb{E}_{s,\mathbf{x}_0,\boldsymbol{\epsilon}\sim N(0,I)}\|A'(s)[(\mathbf{x}_0 - \boldsymbol{\epsilon}) - v_\theta(\mathbf{x}_s(\mathbf{x}_0,\boldsymbol{\epsilon}), A(s))]\|^2. \tag{12}$$

Data is generated by solving the ODE in Equation 7 from $s = 1$ to $s = 0$ with initial condition $\mathbf{x}_1$ sampled from $\mathcal{N}(0, I)$.

*Remark* 4.4. For numerical methods, we define $A'(s)$ at points of non-differentiability as the left-hand derivative $\lim_{s' \to s^-} A'(s')$, ensuring consistency with the piecewise-defined nature of $A(s)$. For piecewise-linear continuous functions $A(s)$ (e.g. the schedule in section 3.2), solving the ODE in Equation 11 with known $\mathbf{x}_0$ and $\boldsymbol{\epsilon}$ with this method using Euler's method or RK4 restores $\mathbf{x}_0$ with zero numerical error.

### 4.4 Model Semantics and Conditional Continuity Equation

TPPs traditionally define distributions over event sequences through a conditional intensity function (Zuo et al., 2020; Mei et al., 2021; Zhang et al., 2020), or a probability density function (PDF) over inter-event times (Shchur et al., 2020). These views allow one to compute the likelihood of a new event given the history, forming the foundation for parametric and neural TPP models. However, in our approach, we adopt a generative perspective: we define a model that implicitly captures the distribution of sequences by learning a transport vector field in latent space. This framework naturally connects to the probabilistic structure of TPPs via the continuity equation.

**Theorem 4.5.** *Let $\mathbf{x}_s = [\mathbf{y}_s, \mathbf{z}_s]$ denote the concatenation of latent variables for observation ($\mathbf{y}$) and prediction ($\mathbf{z}$) windows with initial conditions $\mathbf{y}_1 = \boldsymbol{\epsilon}_{\mathbf{y}}, \mathbf{z}_1 = \boldsymbol{\epsilon}_{\mathbf{z}}$. Suppose the joint variable evolves under the ODE*

$$\frac{d\mathbf{x}_s}{ds} = \mathbf{f}(\mathbf{x}_s, s), \tag{13}$$

*where $\mathbf{f}(\mathbf{x}_s, s)$ is defined in Equation 8 element-wise:*

$$f_i(\mathbf{x}_s, s) = \begin{cases} [A'(s)]_{ii}(\mathbf{y}^{(i)} - \boldsymbol{\epsilon}^{(i)}) & i \in \mathbf{y} \\ [A'(s)]_{ii}[v_\theta(\mathbf{x}_s, A(s))]_i & i \in \mathbf{z}, \end{cases} \tag{14}$$

*and $A(s)$ satisfies the conditions in Assumption 4.1. Then the conditional distribution $p_s(\mathbf{z}_s|\mathbf{y}_s)$ of predictions $\mathbf{z}_s$ conditioned on observations $\mathbf{y}_s$ evolves according to the conditional continuity equation*

$$\partial_s p_s(\mathbf{z}_s|\mathbf{y}_s) + \nabla_{\mathbf{z}_s} \cdot (p_s(\mathbf{z}_s|\mathbf{y}_s) \, \mathbf{f}_{\mathbf{z}}(\mathbf{x}_s, s)) = 0, \tag{15}$$

*where $\mathbf{f}_{\mathbf{z}}$ is the slice of $\mathbf{f}$ in the prediction window, and the boundary condition $p_1(\mathbf{z}_1|\mathbf{y}_1)$ is set to the Gaussian measure. Subsequently, the likelihood of predicted events $\mathbf{z}_0$ conditioned on observed events $\mathbf{y}_0$ is given by*

$$\log p_0(\mathbf{z}_0|\mathbf{y}_0) = \log p_1(\mathbf{z}_1) - \int_0^1 \nabla_{\mathbf{z}} \cdot \mathbf{f}_{\mathbf{z}}(\mathbf{x}_s, s)ds. \tag{16}$$

The proof of the theorem is in Appendix A.5. The conditional continuity equation governs deterministic probability flow in DiTs with arbitrary observation and prediction windows. This equation shows that, at inference time, we learn a distribution of future events given past events in a sequence, which is the canonical objective in TPP modelling.

## 5 Experiments

In this section, we evaluate the performance of ADiff4TPP on prediction tasks for TPP datasets, focusing on two commonly studied tasks: Next event prediction and long horizon prediction. We compare against a selection of baseline methods (details in Appendix C). Note that their numbers are reproduced using the official EasyTPP repository[1].

### 5.1 Next Event Prediction

Table 1: Metrics of next event prediction (RMSE of inter-arrival time / Error Rate of mark). Standard deviations over five seeds are posted below. **Bold** indicates state-of-the-art results in either RMSE or Error Rate. The hyperparameters for ADiff4TPP are: $d_{\text{latent}} = 32, \beta_{\max} = 0.01$.

| Model | Amazon | Retweet | Taxi | Taobao | StackOverflow |
|---|---|---|---|---|---|
| RMTPP | 0.559/70.4% (0.014/0.008) | 26.207/48.4% (5.650/0.030) | 0.351/11.5% (0.042/0.002) | 0.257/56.4% (0.073/0.000) | 1.246/57.6% (0.293/0.002) |
| NHP | 0.640/70.0% (0.002/0.001) | 22.511/46.1% (0.033/0.001) | 0.342/13.0% (0.075/0.007) | 0.168/50.7% (0.098/0.012) | 1.324/73.2% (0.359/0.146) |
| SAHP | 0.517/68.0% (0.008/0.005) | 21.708/46.0% (0.001/0.001) | 0.335/11.9% (0.175/0.016) | 0.154/53.6% (0.083/0.009) | 1.327/57.7% (0.002/0.002) |
| THP | 0.550/65.4% (0.016/0.002) | 26.176/40.5% (0.059/0.001) | 0.375/12.7% (0.065/0.011) | 0.314/54.4% (0.044/0.017) | 1.424/58.0% (0.012/0.013) |
| AttNHP | 0.755/68.1% (0.185/0.011) | 22.296/42.8% (1.135/0.041) | 0.429/14.8% (0.012/0.027) | 0.280/52.9% (0.085/0.019) | 1.350/55.4% (0.018/0.001) |
| IFTTPP | 0.465/**64.9%** (0.001/0.001) | 22.198/40.0% (0.234/0.002) | 0.357/8.56% (0.013/0.0004) | 0.598/44.1% (0.103/0.003) | 1.884/**54.5%** (0.043/0.008) |
| DTPP | 0.619/65.5% (0.104/0.002) | 24.680/40.3% (6.364/0.003) | 0.302/12.10% (0.043/0.012) | 0.587/53.3% (0.031/0.003) | 1.780/60.7% (0.167/0.002) |
| Add-Thin | 0.461/N.A. (0.017/N.A.) | 22.914/N.A. (0.348/N.A.) | 0.368/N.A. (0.015/N.A.) | 0.440/N.A. (0.035/N.A.) | 1.469/N.A. (0.238/N.A.) |
| HYPRO | 0.583/66.2% (0.012/0.004) | 20.562/40.0% (1.633/0.049) | 0.383/13.46% (0.008/0.018) | 0.307/44.9% (0.029/0.004) | 1.417/55.1% (0.253/0.002) |
| ADiff4TPP | **0.407**/67.5% (0.002/0.002) | **17.880/39.3%** (0.051/0.001) | **0.299/8.46%** (0.0002/0.0005) | **0.140/42.6%** (0.054/0.011) | **1.226**/61.3% (0.035/0.011) |

We evaluate the next event prediction of the model by comparing the predicted events against the true events in the test set. Our model predicts the event $\tilde{z}_n = (\tilde{\tau}_n, \tilde{k}_n)$ provided events $z_1, ..., z_{n-1}$ for $n = 2, ..., N$, where $N$ is the length of the event sequence, $\tilde{\tau}_n$ is the predicted inter-arrival time, and $\tilde{k}_n$ is the predicted mark. Similarly to the work of Xue et al. (2024), we evaluate our model by comparing the predicted inter-arrival time against the true inter-arrival time using the root mean square error (RMSE) metric, and by comparing the predicted mark against the true mark using the error rate metric. The results are presented in Table 1. ADiff4TPP outperforms previous methods in next event time prediction across all datasets. Specifically, it improves the Retweet RMSE by 13% (e.g., 17.880 vs. 20.562). On next mark prediction, ADiff4TPP surpasses the previous state-of-the-art on three datasets (i.e., Retweet, Taxi, and Taobao) and yields competitive results on others.

---

[1]https://github.com/ant-research/EasyTemporalPointProcess

## 5.2 Long Horizon Prediction

Next, we evaluate ADiff4TPP for long horizon prediction using the optimal transport distance (OTD) between the predicted events and the true events in the prediction interval using the implementation by Mei et al. (2019). Long horizon predictions are generated by solving the ODE with the vector field provided in Equation 8. The observation window consists of indices $O = \{1, ..., N - h\}$, and the prediction window consists of the index $P = \{N - h + 1, ..., N\}$, where $h$ is the horizon. We refer to Appendix F.2 for the algorithm used for long horizon prediction. The results are summarized in Figure 4 with the complete results deferred to Appendix G. ADiff4TPP achieves state-of-the-art results in long horizon prediction in every dataset over 5, 10, 20, and 30 steps, achieving an average improvement of 14.0% from HYPRO. Furthermore, the performance gap widens as the horizon increases. We attribute the effectiveness of ADiff4TPP on long horizon prediction to the fact that our asynchronous diffusion design enables the DMs to directly model the joint distribution of multiple events while preserving the sequential structure of the data.

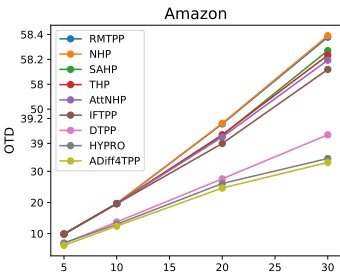 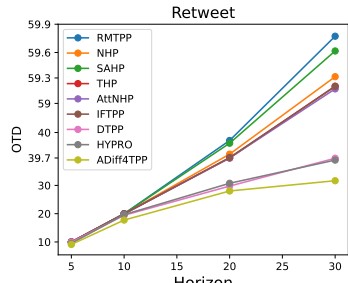 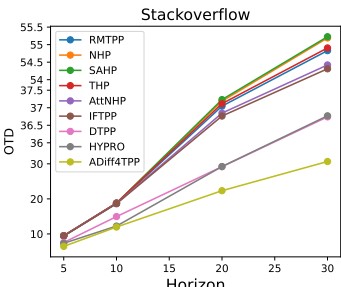

Figure 4: Plots of long horizon prediction conducted on three datasets. ADiff4TPP outperforms the baseline methods in each dataset with respect to the OTD metric. The y-axis is scaled to show the difference between baseline methods.

## 5.3 Ablation Studies

We conduct ablation studies to evaluate the contributions of key design choices in ADiff4TPP. First, we examine the role of the latent space by varying the $\beta$-VAE hyperparameters. We observe that setting $\beta_{\max} \in \{0.01, 0.001\}$ and $d_{latent} \in \{16, 32\}$ yields near-perfect event reconstruction, with better downstream performance when using 32 dimensions except on the StackOverflow dataset. Full reconstruction results are reported in Appendix E. We also compare three noise schedules: our proposed asynchronous schedule, a synchronous (rectified flow) variant, and a disjoint (autoregressive-like) schedule. The schedules are detailed in Appendix H, and in all settings, we keep the architecture, latent representation, and masking identical. As shown in Table 2, the asynchronous schedule consistently outperforms the alternatives, confirming its effectiveness for event generation. Additionally, we study the impact of masking in the DiT transformer. Without masking, all events—including future ones—are visible to each token during attention, regardless of their noise levels. As shown in Row 3 of Table 2, masked models outperform their unmasked counterparts, supporting the idea that masking helps isolate informative signals during generation.

Finally, to evaluate whether the VAE formulation is necessary, we compare the proposed $\beta$-VAE representation with two alternative event encoding strategies: (i) a deterministic autoencoder (AE) with the same architecture but without KL regularization, and (ii) a standard embedding consisting of a learnable mark embedding and a Time2Vec encoding (Kazemi et al., 2019) for inter-event time. All models use the same latent dimension and noise schedule. Table 8 reports the reconstruction performance and the empirical KL divergence of the learned latent distribution from a unit Gaussian, and Table 9 reports the long horizon prediction performance. While all methods achieve comparable reconstruction accuracy on the simpler Amazon dataset, the deterministic AE and standard embeddings exhibit substantially larger deviations from the Gaussian prior. When used within the DM, these representations lead to worse long-horizon forecasting performance. This suggests that the benefit of the VAE is not only reconstruction accuracy, but also the geometry of the latent space. Further explanation is provided in Appendix I. The variational regularization encourages a smooth latent distribution

Table 2: Ablation Study Results on Next Event prediction. We compared ADiff4TPP with different choices of noise schedules as well as different latent dimensions and $\beta_{max}$ for training the VAE. **Bold** indicates state-of-the-art results in either RMSE or Error Rate.

| Model | Amazon | Retweet | Taobao | StackOverflow |
|---|---|---|---|---|
| Disjoint Diffusion | 0.492/69.9% | 18.813/47.3% | 0.506/48.2% | 1.123/65.7% |
| $d_{\text{latent}} = 32, \beta_{\text{max}} = 0.01$ | (0.009/0.006) | (0.096/0.008) | (0.069/0.027) | (0.042/0.002) |
| Disjoint Diffusion | 0.483/69.5% | 20.054/49.4% | 0.461/49.3% | 1.544/69.4% |
| $d_{\text{latent}} = 32, \beta_{\text{max}} = 0.001$ | (0.007/0.001) | (0.099/0.007) | (0.054/0.008) | (0.038/0.008) |
| Synchronized Diffusion | 0.432/68.5% | 18.358/45.8% | 0.436/45.7% | 1.285/61.7% |
| $d_{\text{latent}} = 32, \beta_{\text{max}} = 0.01$ | (0.002/0.003) | (0.103/0.003) | (0.139/0.022) | (0.041/0.004) |
| Synchronized Diffusion | 0.424/68.0% | 18.085/45.4% | 0.422/46.9% | 1.336/62.3% |
| $d_{\text{latent}} = 32, \beta_{\text{max}} = 0.001$ | (0.001/0.002) | (0.062/0.004) | (0.063/0.010) | (0.011/0.005) |
| Unmasked Diffusion | 0.408/68.0% | 19.513/45.6% | 0.221/54.3% | 1.249/60.6% |
| $d_{\text{latent}} = 32, \beta_{\text{max}} = 0.01$ | (0.001/0.002) | (0.964/0.024) | (0.028/0.030) | (0.019/0.014) |
| Unmasked Diffusion | 0.495/71.8% | 19.803/46.0% | 0.174/53.1% | 1.094/59.9% |
| $d_{\text{latent}} = 32, \beta_{\text{max}} = 0.001$ | (0.013/0.057) | (0.269/0.036) | (0.025/0.023) | (0.015/0.019) |
| ADiff4TPP | 0.440/68.7% | 19.383/53.2% | 0.354/54.9% | **1.090/57.7%** |
| $d_{\text{latent}} = 16, \beta_{\text{max}} = 0.01$ | (0.004/0.002) | (0.041/0.003) | (0.155/0.015) | (0.012/0.006) |
| ADiff4TPP | 0.421/68.5% | 17.857/41.6% | 0.326/46.2% | 1.310/59.8% |
| $d_{\text{latent}} = 16, \beta_{\text{max}} = 0.001$ | (0.001/0.002) | (0.030/0.0003) | (0.017/0.03) | (0.058/0.012) |
| ADiff4TPP | **0.407**/67.5% | 17.880/**39.3%** | **0.140/42.6%** | 1.226/61.3% |
| $d_{\text{latent}} = 32, \beta_{\text{max}} = 0.01$ | (0.002/0.002) | (0.051/0.001) | (0.054/0.011) | (0.035/0.011) |
| ADiff4TPP | 0.436/**67.4%** | **17.271**/39.4% | 0.177/44.1% | 1.169/63.5% |
| $d_{\text{latent}} = 32, \beta_{\text{max}} = 0.001$ | (0.003/0.0003) | (0.010/0.002) | (0.022/0.010) | (0.078/0.015) |

aligned with the Gaussian noise assumption used by diffusion models, leading to more stable generation and improved long-horizon predictions.

The ablation results highlight the role of the noise schedule and masking strategy in ADiff4TPP. In particular, synchronous noise schedules consistently underperform asynchronous schedules across datasets. This suggests that treating all events as equally uncertain throughout the generation process limits the model's ability to exploit temporal structure, especially when forecasting multiple future events jointly. At the other extreme, disjoint (fully autoregressive) schedules also underperform, indicating that sequentially denoising one event at a time sacrifices the benefits of joint modelling and increases error propagation. The proposed asynchronous schedule strikes a balance between these two regimes by allowing earlier events in the prediction window to stabilize before later ones, while still enabling joint, non-autoregressive generation.

## 6 Related Works

Following the work of Rombach et al. (2022), performing diffusion in latent spaces has been of high interest to the DM community. Text-to-image models like Stable Diffusion (Esser et al., 2024) have continued to use latent spaces to generate high resolution realistic images. The work of TabSyn (Zhang et al., 2024) extends latent diffusion models to the generation of tabular data. The authors trained a $\beta$-VAE to transform each table row into latent vectors and then trained a score-based diffusion model (Song et al., 2021) to generate latent rows. TabSyn excels at filling in missing data and creating high-quality synthetic data.

DMs with asynchronous noise schedules are a recent area of interest. Groupwise diffusion models (Lee et al., 2023) divide an image into groups and diffuse each group separately to perform cascaded generation of images

in pixel and frequency space. In the work of Diffusion Forcing (Chen et al., 2024), each time step is assigned a randomly sampled noise level, and the diffusion model is trained to denoise data according to arbitrary noise schedules. This is applied to video generation, motion planning, and robotics.

Add-Thin (Lüdke et al., 2023) is one of the first papers to use DMs for TPPs by learning the intensity function. The reverse diffusion process is motivated by algorithms such as thinning and superposition that are used in prior intensity-based TPP approaches. Event Flow (Kerrigan et al., 2024) extends DMs to TPPs by conditioning on preceding events. The method predicts future events in a prediction horizon through a single denoising process rather than by cascadedly performing next event predictions.

Recent advances in generative modelling for point processes have expanded to include diverse neural and diffusion-based approaches. The work of Yuan et al. (2023) introduces a framework for jointly modelling temporal and spatial point process data using a DM, addressing conditional independence between temporal and spatial dynamics. In contrast, Lüdke et al. (2024) models point processes as unordered sets via point cloud denoising, enabling permutation-invariant generation and direct likelihood evaluation.

## 7 Conclusion

In this paper, we studied the problem of modelling TPPs using asynchronous diffusion-based generative models. Our goal was to address several challenges inherent to TPPs, including mixed continuous–categorical event representations, conditioning on partial observation histories of variable length, and efficient long-horizon forecasting. To this end, we introduced ADiff4TPP, which assigns different noise scales to latent representations of different events within a sequence. This design allows the model to prioritize near-future events while progressively generating more distant ones, providing an efficient alternative to strictly autoregressive generation. The proposed asynchronous noise schedule further enables flexible control over the prediction window, allowing both short- and long-horizon forecasting within a unified framework. When trained using a CFM objective with a DiT backbone, ADiff4TPP consistently outperforms baselines across five benchmark datasets in both short- and long-horizon prediction tasks.

More broadly, this work investigated asynchronous diffusion as a modelling principle for sequential data, and instantiated it concretely through a CFM framework with a matrix-valued, piecewise-linear noise schedule. We showed how such schedules induce a temporally ordered denoising process, in which earlier elements of a sequence are reconstructed before later ones, while still supporting joint, non-autoregressive prediction over future windows. To make this construction well defined, we addressed technical challenges arising from non-invertibility and piecewise differentiability of the noise schedule, and derived conditions under which the resulting flows remain valid. Applied to marked TPPs in a learned latent space, this approach yields consistent empirical gains over existing methods. Together, these results suggest that asynchronous diffusion provides a practical and theoretically grounded alternative to synchronous or strictly autoregressive approaches for TPP modelling.

## 8 Limitations and Future Work

While our method enables scalable and flexible modelling of marked TPPs, it inherits several limitations common to transformer-based generative architectures. First, the model's context window, that is, the number of past events the model can condition on, is constrained by the maximum sequence length encountered during training. This can limit performance when the test-time event sequences are significantly longer than those seen during training. Future work can explore data augmentation strategies that synthetically extend event histories or interpolate between partial sequences, allowing the model to train with a more diverse range of context lengths.

While our approach effectively models event sequences using asynchronous noise schedules, it primarily relies on context from preceding events in the sequence. However, many real-world applications, such as financial forecasting, social media trend prediction, and medical event modelling, can benefit from external knowledge sources. Inspired by recent advancements in text-to-image generation using latent diffusion models (Rombach et al., 2022; Esser et al., 2024), a promising direction for future work is *text-to-TPP generation*, where event

sequences are conditioned not only on past events but also on external context. Large pretrained language models (e.g., GPT-4 (OpenAI, 2023)) offer extensive world knowledge and reasoning capabilities that can significantly enhance short and long horizon predictions when integrated with TPPs. These models can serve as external context providers, enabling more accurate, context-aware event sequence forecasting. This approach could bridge the gap between unstructured textual data and structured TPPs, opening up new avenues for more interpretable, data-efficient, and semantically grounded generative models. Concurrent with our work, language-modelled asynchronous time series (LAST SToP (Gupta et al., 2025)) has made an early step in this direction by introducing stochastic soft prompting to leverage external data for modelling asynchronous time series.

Finally, our current study focuses exclusively on TPPs, and does not evaluate the applicability of asynchronous diffusion to broader classes of sequential data. While our formulation is naturally suited to event sequences with mixed continuous–categorical attributes, many real-world sequential modelling problems, such as time series forecasting, also exhibit heterogeneous data types and temporal dependencies. Extending our framework to these more general settings is a promising direction for future work.

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

# A    Proofs on the Flow Matching Objective

## A.1    Physical Interpretability of the Flow

*Remark* A.1. Assume the noise schedule $A(s)$ satisfies conditions (1)–(3) in Assumption 4.1. The evolution of the distribution at flow time $s$ due to the flow $\psi_s(\cdot|\cdot)$ is:

$$p_s(\mathbf{x}_s|\mathbf{x}_0) = \mathcal{N}\left(\mathbf{x}_s; A(s)\mathbf{x}_0, (I - A(s))^T(I - A(s))\right). \tag{17}$$

The expectation term $A(s)\mathbf{x}_0$ governs how much of the clean data $\mathbf{x}_0$ is retained as the distribution evolves. The covariance term $(I - A(s))^T(I - A(s))$ reflects the increasing uncertainty introduced by Gaussian noise.

The following properties must be satisfied for the flow to have physical interpretability. These properties are observed, not only in flow matching, but also in score-based diffusion models (Song et al., 2021) and rectified flows (Liu et al., 2023).

Firstly, at $s = 0$, the clean data $\mathbf{x}_0$ is completely retained, and the covariance term is at a minimum. Consequently, at $s = 1$, the clean data $\mathbf{x}_0$ is no longer retained, and the covariance term is at a maximum. To align with the dissipative nature of diffusion processes (Planck, 1926), it suffices for the mean term to be monotone non-increasing with $s$, and the covariance to be monotone non-decreasing with $s$. This ensures that the clean data loses retention and the uncertainty increases.

Lastly, the flow needs to be continuous in order for us to model it as an ODE. It suffices for us to show that the conditions met by $A(s)$ satisfy physical interpretability.

- It is sufficient for $A(s)$ to satisfy the boundary conditions shown in condition (1) as $\mathbb{E}[\mathbf{x}_0|\mathbf{x}_0] = \mathbf{x}_0, \mathbb{E}[\mathbf{x}_1|\mathbf{x}_0] = 0$ is a sufficient condition for the clean data $\mathbf{x}_0$ to be obtainable at $s = 0$ and unobtainable at $s = 1$ due to the evolution of the distribution.

- Conditions (2) and (3) are sufficient to show that $\mathbb{E}[\mathbf{x}_0^T\mathbf{x}_s|\mathbf{x}_0] = \mathbf{x}_0^T A(s)\mathbf{x}_0$ is non-negative and non-increasing with respect to $s$ (Löwner, 1934). As a result, $A(s)$ does not amplify any components of $\mathbf{x}_0$. They also show that $\text{Cov}[\mathbf{x}_s|\mathbf{x}_0] = (I - A(s))^T(I - A(s))$ is non-negative and non-decreasing with respect to $s$.

As a result, the conditions show that $A(s)$ aligns with the dissipative nature of diffusion processes.

## A.2    Validity of Inverse Flows

**Definition A.2.** The inverse flow $\psi_s^{-1}(\mathbf{x}_s|\boldsymbol{\epsilon})$ is considered to be "well-posed" if all partially diffused elements of $\mathbf{x}_s$ at flow time $s$ can be reconstructed. Instead of returning a deterministic variable $\mathbf{x}_0$, it returns a distribution of values $\mathbf{x}_0$ can take based on partial observations. Sufficiently, the flow $\psi(\mathbf{x}_s|\boldsymbol{\epsilon})$ pushed the distribution of $\psi_s^{-1}(\mathbf{x}_s|\boldsymbol{\epsilon})$ back to the original distribution of $\mathbf{x}_s$.

We observe that $\mathbb{E}[\psi_s^{-1}(\mathbf{x}_s|\boldsymbol{\epsilon})|\boldsymbol{\epsilon}] = A(s)^\dagger A(s)\mathbf{x}_0$, where $A(s)^\dagger A(s)$ acts as a projection matrix, projecting $\mathbf{x}_0$ onto the rank space of $A(s)$. The components of $\mathbf{x}_s$ lying in the null space of $A(s)$ are replaced by Gaussian noise during the process.

By aligning the distribution of the generated samples with the forward flow $\psi_s(\cdot|\cdot)$, we aim to recover the distribution of $\mathbf{x}_s$. Particularly, we find that $\mathbb{E}[\psi_s(\psi_s^{-1}(\mathbf{x}_s|\boldsymbol{\epsilon})|\boldsymbol{\epsilon})|\boldsymbol{\epsilon}] = A(s)A(s)^\dagger A(s)\mathbf{x}_0 = A(s)\mathbf{x}_0$, which demonstrates consistency with the projection behavior of $A(s)$. This observation motivates the need to rigorously establish that $p_s(\psi(\psi_s^{-1}(\mathbf{x}_s|\boldsymbol{\epsilon})|\boldsymbol{\epsilon})|\boldsymbol{\epsilon}) = p_s(\mathbf{x}_s|\boldsymbol{\epsilon})$, ensuring that the reconstructed distribution matches the original one.

**Lemma A.3** (Validity of $\psi_s^{-1}$). *The inverse flow $\psi_s^{-1}(\mathbf{x}_s|\boldsymbol{\epsilon}) = A(s)^\dagger(\mathbf{x}_s - \boldsymbol{\epsilon}) + \boldsymbol{\epsilon}$ is well-posed and reconstructs all of the partially-diffused elements of $\mathbf{x}_s$ if $A(s)$ satisfies the conditions in Assumption 4.1 for all $s \in [0, 1]$. Formally, $\psi_s^{-1}(\mathbf{x}_s|\boldsymbol{\epsilon})$ is left-invertible i.e. $\psi_s(\psi_s^{-1}(\mathbf{x}_s|\boldsymbol{\epsilon})|\boldsymbol{\epsilon}) = \mathbf{x}_s$.*

*Proof.* To prove that our choice of $\psi_s^{-1}$ is valid, we need to show that $\psi_s(\psi_s^{-1}(\mathbf{x}_s|\boldsymbol{\epsilon})|\boldsymbol{\epsilon}) = \mathbf{x}_s$.

$$
\begin{aligned}
\psi_s(\psi_s^{-1}(\mathbf{x}_s|\boldsymbol{\epsilon})|\boldsymbol{\epsilon}) &= A(s)\psi_s^{-1}(\mathbf{x}_s|\boldsymbol{\epsilon}) + (I - A(s))\boldsymbol{\epsilon} \\
&= A(s)A(s)^{\dagger}(\mathbf{x}_s - \boldsymbol{\epsilon}) + A(s)\boldsymbol{\epsilon} + (I - A(s))\boldsymbol{\epsilon} \\
&= A(s)A(s)^{\dagger}\mathbf{x}_s + (I - A(s)A(s)^{\dagger})\boldsymbol{\epsilon}.
\end{aligned}
\tag{18}
$$

We expand $\mathbf{x}_0$ and use the property of the Moore-Penrose Pseudo-Inverse that $A(s)A(s)^{\dagger}A(s) = A(s)$ to restore $\mathbf{x}_s$.

$$
\begin{aligned}
\psi_s(\psi_s^{-1}(\mathbf{x}_s|\boldsymbol{\epsilon})|\boldsymbol{\epsilon}) &= A(s)A(s)^{\dagger}[A(s)\mathbf{x}_0 + (I - A(s))\boldsymbol{\epsilon}] + (I - A(s)A(s)^{\dagger})\boldsymbol{\epsilon} \\
&= A(s)\mathbf{x}_0 + A(s)A(s)^{\dagger}\boldsymbol{\epsilon} - A(s)\boldsymbol{\epsilon} + (I - A(s)A(s)^{\dagger})\boldsymbol{\epsilon} \\
&= A(s)\mathbf{x}_0 + (I - A(s))\boldsymbol{\epsilon} \\
&= \mathbf{x}_s.
\end{aligned}
\tag{19}
$$

This completes the proof. $\qquad\square$

### A.3 Equivalence of Flow Matching Objectives

To show that the flow matching objectives in Equations 10 and 11 are equivalent, we first prove the following property about projection matrices.

**Lemma A.4.** *Let $P(s) \in H^1([0,1]; \mathbb{R}^{n \times n})$ be a family of orthogonal projection matrices i.e. $P(s)^2 = P(s) = P(s)^T$. Then its derivative $P'(s)$ exists in the weak sense and is orthogonal to $P(s)$ for every $s \in (0,1)$.*

*Proof.* We start from $P(s)^2 = P(s)$ and take the weak derivative of both sides. Note that the product rule holds in the weak sense.

$$
P(s)P'(s) + P'(s)P(s) = P'(s).
\tag{20}
$$

We then left- and right- multiply both sides by $P(s)$ and use $P(s)^2 = P(s)$ to obtain

$$
P(s)P'(s)P(s) + P(s)P'(s)P(s) = P(s)P'(s)P(s),
\tag{21}
$$

which simplifies to $P(s)P'(s)P(s) = 0$. Since $P(s)$ is symmetric for all $s \in [0,1]$, this means $P'(s)$ is symmetric for all $s \in (0,1)$. We exploit this property to show that

$$
P(s)P'(s)P(s) = 0 \implies P'(s)P(s) = P(s)P'(s) = 0 \text{ for all } s \in (0,1),
\tag{22}
$$

which is a well-known result in linear algebra. $\qquad\square$

**Proposition A.5.** *Let $A(s) \in H^1([0,1]; \mathbb{R}^{n \times n})$ satisfy the conditions of Assumption 4.1. Then the vector fields $A'(s)A(s)^{\dagger}[\mathbf{x}_s - \boldsymbol{\epsilon}]$ and $A'(s)[\mathbf{x}_0 - \boldsymbol{\epsilon}]$ are equivalent.*

*Proof.* Expanding $\mathbf{x}_s$ gives

$$
\begin{aligned}
&A'(s)A(s)^{\dagger}[\mathbf{x}_s - \boldsymbol{\epsilon}] \\
&= A'(s)A(s)^{\dagger}[A(s)\mathbf{x}_0 + (1 - A(s))\boldsymbol{\epsilon} - \boldsymbol{\epsilon}] \\
&= A'(s)A(s)^{\dagger}A(s)[\mathbf{x}_0 - \boldsymbol{\epsilon}].
\end{aligned}
\tag{23}
$$

To simplify this expression, we need to interpret $A'(s)A(s)^\dagger A(s)$, where $A'(s)$ is the weak derivative of $A(s)$. Let $\xi(s)$ be a scalar function that is zero outside of $[0, 1]$.

$$\int_0^1 A'(s)A(s)^\dagger A(s)\xi(s)ds$$
$$= -\int_0^1 A(s)\frac{d}{ds}[A(s)^\dagger A(s)]\xi(s)ds - \int_0^1 A(s)A(s)^\dagger A(s)\xi'(s)ds$$
$$= -\int_0^1 A(s)\frac{d}{ds}[A(s)^\dagger A(s)]\xi(s)ds - \int_0^1 A(s)\xi'(s)ds \tag{24}$$
$$= -\int_0^1 A(s)\frac{d}{ds}[A(s)^\dagger A(s)]\xi(s)ds + \int_0^1 A'(s)\xi(s)ds.$$

By definition, $P(s) = A(s)^\dagger A(s)$ is a projection matrix that projects vectors to the row space of $A(s)$. Using Lemma A.4 and the fact that $A(s) = A(s)P(s)$, we conclude that

$$A(s)P'(s) = A(s)P(s)P'(s) = 0 \text{ for all } s \in (0, 1), \tag{25}$$

which follows from $P(s)P'(s) = 0$. Finally, since $\xi(s)$ vanishes at $s \in \{0, 1\}$, we conclude that

$$\int_0^1 A(s)\frac{d}{ds}[A(s)^\dagger A(s)]\xi(s)ds = 0. \tag{26}$$

Since this holds for all $\xi(s)$ with compact support, it shows that $A'(s)A(s)^\dagger A(s) = A'(s)$, which completes the proof. $\square$

### A.4 Essential Properties of Flow Matching

**Lemma 4.2.** We argue that the properties of the conditional vector field $u_s(\mathbf{x}_s|\boldsymbol{\epsilon})$ still hold even if we relax the invertible assumptions of $A(s)$ as long as the conditions of Assumption 4.1 hold. We characterize the essential properties of flow matching as follows:

1. The conditional vector field $u_s(\mathbf{x}_s|\boldsymbol{\epsilon})$ can be derived in closed form.

2. The marginal vector field $u_s(\mathbf{x}_s)$ can be constructed by marginalizing over the conditional vector field $u_s(\mathbf{x}_s|\boldsymbol{\epsilon})$ that generates the flow. In other words: $\mathbb{E}_{\boldsymbol{\epsilon}\sim\mathcal{N}(0,I)}[\frac{p_s(\mathbf{x}_s|\boldsymbol{\epsilon})}{p_s(\mathbf{x}_s)}u_s(\mathbf{x}_s|\boldsymbol{\epsilon})] = u_s(\mathbf{x}_s)$.

3. The training objective for the marginal vector field $A'(s)v_\theta(\mathbf{x}_s, A(s))$ can be expressed in terms of the conditional vector field $u_s(\mathbf{x}_s|\boldsymbol{\epsilon})$.

The flow $\psi(\mathbf{x}_s|\boldsymbol{\epsilon})$ in Equation 5 governed by a matrix-valued coefficient $A(s) \in H^1([0,1];\mathbb{R}^{n\times n})$ remains valid as long as the following conditions are met:

1. $A(s)$ satisfies the boundary conditions: $A(0) = I, A(1) = 0$.

2. $A(s)$ is positive semi-definite for all $s \in [0, 1]$

3. $A(s)$ is monotone non-increasing in the Löwner order i.e. $A(s) \preceq A(s')$ if $s \geq s'$.

4. $A(s)$ is continuous for all $s \in [0, 1]$.

*Proof.* We will proceed to show that these conditions are still satisfied. We can expand the conditional vector field as $u_s(\mathbf{x}_s|\boldsymbol{\epsilon}) := A'(s)\nu_s(\mathbf{x}_s|\boldsymbol{\epsilon})$ and $u_s(\mathbf{x}_s) := A'(s)\nu_s(\mathbf{x}_s)$. While condition (1) is straightforward, the demonstration of conditions (2) and (3) is remarkably identical to Esser et al. (2024, Appendix B.1).

1. This condition is trivially satisfied as shown in Proposition A.5.

2. The continuity equation provides a sufficient condition to determine if $u_s(\mathbf{x}_s)$ models the evolution of the distribution $p_s(\mathbf{x}_s)$:

$$\frac{d}{ds}p_s(\mathbf{x}_s) = -\nabla \cdot [p_s(\mathbf{x}_s)u_s(\mathbf{x}_s)] = -\nabla \cdot [p_s(\mathbf{x}_s)A'(s)\nu_s(\mathbf{x}_s)]. \tag{27}$$

This equation can be expanded to show that the marginal vector field $u_s(\mathbf{x}_s|\boldsymbol{\epsilon})$ models the evolution of the marginal distribution $p_s(\mathbf{x}_s|\boldsymbol{\epsilon})$:

$$\begin{aligned}
\nabla \cdot [p_s(\mathbf{x}_s)A'(s)\nu_s(\mathbf{x}_s)] &= \nabla \cdot [\mathbb{E}_{\boldsymbol{\epsilon}\sim\mathcal{N}(0,I)}[A'(s)\nu_s(\mathbf{x}_s|\boldsymbol{\epsilon})\frac{p_s(\mathbf{x}_s|\boldsymbol{\epsilon})}{p_s(\mathbf{x}_s)}p_s(\mathbf{x}_s)]] \\
&= \mathbb{E}_{\boldsymbol{\epsilon}\sim\mathcal{N}(0,I)}[\nabla \cdot [A'(s)\nu_s(\mathbf{x}_s|\boldsymbol{\epsilon})\frac{p_s(\mathbf{x}_s|\boldsymbol{\epsilon})}{p_s(\mathbf{x}_s)}p_s(\mathbf{x}_s)]] \\
&= \mathbb{E}_{\boldsymbol{\epsilon}\sim\mathcal{N}(0,I)}[-\frac{d}{ds}p_s(\mathbf{x}_s|\boldsymbol{\epsilon})] \\
&= -\frac{d}{ds}p_s(\mathbf{x}_s)
\end{aligned} \tag{28}$$

3. To prove this condition, it is first sufficient to show that $\langle A'(s)v_\theta(\mathbf{x}_s, A(s)), u_s(\mathbf{x}_s)\rangle$ can be constructed from $\langle A'(s)v_\theta(\mathbf{x}_s, A(s)), u_s(\mathbf{x}_s|\boldsymbol{\epsilon})\rangle$.

$$\begin{aligned}
&\mathbb{E}_{s,p_s(\mathbf{x}_s|\boldsymbol{\epsilon}),p(\boldsymbol{\epsilon})}\langle A'(s)v_\theta(\mathbf{x}_s, A(s)), A'(s)\nu_s(\mathbf{x}_s|\boldsymbol{\epsilon})\rangle \\
&= \iint \langle A'(s)v_\theta(\mathbf{x}_s, A(s)), A'(s)\nu_s(\mathbf{x}_s|\boldsymbol{\epsilon})\rangle p_s(\mathbf{x}_s|\boldsymbol{\epsilon})p(\boldsymbol{\epsilon})d\mathbf{x}_s d\boldsymbol{\epsilon} \\
&= \int \langle A'(s)v_\theta(\mathbf{x}_s, A(s)), A'(s)\int \frac{p_s(\mathbf{x}_s|\boldsymbol{\epsilon})p(\boldsymbol{\epsilon})}{p_s(\mathbf{x}_s)}\nu_s(\mathbf{x}_s|\boldsymbol{\epsilon})d\boldsymbol{\epsilon}\rangle p_s(\mathbf{x}_s)d\mathbf{x}_s \\
&= \int \langle A'(s)v_\theta(\mathbf{x}_s, A(s)), A'(s)\nu_s(\mathbf{x}_s)\rangle p_s(\mathbf{x}_s)d\mathbf{x}_s \\
&= \mathbb{E}_{s,p_s(\mathbf{x}_s)}\langle A'(s)v_\theta(\mathbf{x}_s, A(s)), A'(s)\nu_s(\mathbf{x}_s)\rangle.
\end{aligned} \tag{29}$$

We can then derive the conditional flow matching objective, $\mathcal{L}_{CFM}(\theta)$, from the flow matching objective, $\mathcal{L}_{FM}(\theta)$.

$$\begin{aligned}
\mathcal{L}_{FM}(\theta) &= \mathbb{E}_{s,p_s(\mathbf{x}_s)}\|A'(s)[v_\theta(\mathbf{x}_s, A(s)) - u_s(\mathbf{x}_s)]\|^2 \\
&= \mathbb{E}_{s,p_s(\mathbf{x}_s)}\|A'(s)v_\theta(\mathbf{x}_s, A(s))\|^2 - 2\mathbb{E}_{s,p_s(\mathbf{x}_s)}\langle A'(s)v_\theta(\mathbf{x}_s, A(s)), A'(s)u_s(\mathbf{x}_s)\rangle + c \\
&= \mathbb{E}_{s,p_s(\mathbf{x}_s)}\|A'(s)v_\theta(\mathbf{x}_s, A(s))\|^2 - 2\mathbb{E}_{s,p_s(\mathbf{x}_s|\boldsymbol{\epsilon}),p(\boldsymbol{\epsilon})}\langle A'(s)v_\theta(\mathbf{x}_s, A(s)), A'(s)u_s(\mathbf{x}_s|\boldsymbol{\epsilon})\rangle + c \\
&= \mathbb{E}_{s,p_s(\mathbf{x}_s|\boldsymbol{\epsilon}),p(\boldsymbol{\epsilon})}\|A'(s)[v_\theta(\mathbf{x}_s, A(s)) - u_s(\mathbf{x}_s|\boldsymbol{\epsilon})]\|^2 + c' \\
&= \mathcal{L}_{CFM}(\theta) + c',
\end{aligned} \tag{30}$$

where $c, c'$ are constants that do not depend on $\theta$.

$\square$

## A.5 Conditional Continuity Equation

**Theorem 4.5** Let $\mathbf{x}_s = [\mathbf{y}_s, \mathbf{z}_s]$ denote the concatenation of latent variables for observation ($\mathbf{y}$) and prediction ($\mathbf{z}$) windows with initial conditions $\mathbf{y}_1 = \boldsymbol{\epsilon}_\mathbf{y}, \mathbf{z}_1 = \boldsymbol{\epsilon}_\mathbf{z}$. Suppose the joint variable evolves under the ODE

$$\frac{d\mathbf{x}_s}{ds} = \mathbf{f}(\mathbf{x}_s, s), \tag{31}$$

where $\mathbf{f}(\mathbf{x}_s, s)$ is defined in Equation 8 element-wise:

$$f_i(\mathbf{x}_s, s) = \begin{cases} [A'(s)]_{ii}(\mathbf{y}^{(i)} - \boldsymbol{\epsilon}^{(i)}) & i \in \mathbf{y} \\ [A'(s)]_{ii}[v_\theta(\mathbf{x}_s, A(s))]_i & i \in \mathbf{z}, \end{cases} \tag{32}$$

and $A(s)$ satisfies the conditions in Assumption 4.1. Then the likelihood of predicted events $\mathbf{y}_0$ conditioned on observed events $\mathbf{z}_0$ is given by

$$\log p_0(\mathbf{z}_0|\mathbf{y}_0) = \log p_1(\mathbf{z}_1) - \int_0^1 \nabla_{\mathbf{z}} \cdot \mathbf{f}_{\mathbf{z}}(\mathbf{x}_s, s)ds, \tag{33}$$

where $\mathbf{f}_{\mathbf{z}}$ is the slice of $\mathbf{f}$ in the prediction window.

*Proof.* We denote $\mathbf{f}_{\mathbf{y}}$ and $\mathbf{f}_{\mathbf{z}}$ as slices of $\mathbf{f}$ in the observation and prediction windows, respectively. The time-evolving joint density $p_s(\mathbf{x}_s) = p_s(\mathbf{y}_s, \mathbf{z}_s)$ is governed by the continuity equation derived in the proof of Lemma 4.2. Performing a change of variables $p_s(\mathbf{x}_s) \rightarrow \log p_s(\mathbf{x}_s)$ gives us the following equation, well known in the works of normalizing flows (Grathwohl et al., 2018; Bol'shev, 1967):

$$\log p_0(\mathbf{y}_0, \mathbf{z}_0) = \log p_1(\mathbf{y}_1, \mathbf{z}_1) - \int_0^1 \nabla \cdot \mathbf{f}(\mathbf{x}_s, s)ds. \tag{34}$$

Applying the chain rule to the log of the joint density

$$\log p_s(\mathbf{y}_s, \mathbf{z}_s) = \log p_s(\mathbf{z}_s|\mathbf{y}_s) + \log p_s(\mathbf{y}_s), \tag{35}$$

we write

$$\log p_0(\mathbf{z}_0|\mathbf{y}_0) = \log p_1(\mathbf{z}_1|\mathbf{y}_1) - \int_0^1 \nabla \cdot \mathbf{f}(\mathbf{x}_s, s)ds - [\log p_0(\mathbf{y}_0) - \log p_1(\mathbf{y}_1)]. \tag{36}$$

Since $\mathbf{f}(\mathbf{x}_s, s)$ at the observation window $\mathbf{y}$ is independent of the variables in the prediction window $\mathbf{z}$, the evolution of $\log p_s(\mathbf{z}_s)$ is also governed by the change of variables formula:

$$\log p_0(\mathbf{y}_0) = \log p_1(\mathbf{y}_1) - \int_0^1 \nabla_{\mathbf{y}} \cdot \mathbf{f}_{\mathbf{y}}(\mathbf{y}_s, s)ds. \tag{37}$$

Substituting this gives

$$\begin{aligned} \log p_0(\mathbf{z}_0|\mathbf{y}_0) &= \log p_1(\mathbf{z}_1|\mathbf{y}_1) - \int_0^1 \nabla \cdot \mathbf{f}(\mathbf{x}_s, s) - \nabla_{\mathbf{y}} \cdot \mathbf{f}_{\mathbf{y}}(\mathbf{y}_s, s)ds \\ &= \log p_1(\mathbf{z}_1|\mathbf{y}_1) - \int_0^1 \nabla_{\mathbf{z}} \cdot \mathbf{f}_{\mathbf{z}}(\mathbf{x}_s, s)ds. \end{aligned} \tag{38}$$

Since $\mathbf{z}_1$ is sampled from $\mathcal{N}(0, I)$, it is independent of $y_1$, thus giving us $\log p_1(\mathbf{z}_1|\mathbf{y}_1) = \log p_1(\mathbf{z}_1)$, which gives the desired likelihood. $\square$

## B An Invertible Noise Schedule

In this section, we will consider analogs to the asynchronous flow matching by strictly enforcing invertibility in the flow. Let $\sigma_{\min} \in (0, 1)$ be a small constant. Consider a family of noise schedules $A_{\sigma_{\min}}(s) \in H^1([0,1]; \mathbb{R}^{n \times n})$ that satisfies the following conditions for $\sigma_{\min}$, modified from Assumption 4.1:

1. $A_{\sigma_{\min}}(s)$ satisfies the boundary conditions: $A_{\sigma_{\min}}(0) = I, A_{\sigma_{\min}}(1) = \sigma_{\min}I$.

2. $A_{\sigma_{\min}}(s)$ is positive definite for all $s \in [0, 1]$

3. $A_{\sigma_{\min}}(s)$ is monotone non-increasing in the Löwner order i.e. $A_{\sigma_{\min}}(s) \preceq A_{\sigma_{\min}}(s')$ if $s \geq s'$.

4. $A_{\sigma_{\min}}(s)$ is continuous for all $s \in [0, 1]$.

Without loss of generality, we can let $A_{\sigma_{\min}}(s)$ be continuous with respect to $\sigma_{\min} \in [0, 1]$ in the $H^1([0, 1]; \mathbb{R}^{n \times n})$ norm i.e. for all $\epsilon > 0$, there exists a $\delta > 0$ such that

$$
\begin{aligned}
|\sigma_{\min} - \sigma'_{\min}| < \delta \implies &\|A_{\sigma_{\min}}(s) - A_{\sigma'_{\min}}(s)\|_{H^1([0,1];\mathbb{R}^{n \times n})} \\
&= \|A_{\sigma_{\min}}(s) - A_{\sigma'_{\min}}(s)\|_{L^2([0,1];\mathbb{R}^{n \times n})} + \|A'_{\sigma_{\min}}(s) - A'_{\sigma'_{\min}}(s)\|_{L^2([0,1];\mathbb{R}^{n \times n})} < \epsilon.
\end{aligned}
$$

Note that the $L^2([0, 1]; \mathbb{R}^{n \times n})$ norm is defined as

$$
\|A(s)\|^2_{L^2([0,1];\mathbb{R}^{n \times n})} = \sum_{i=1}^{n} \sum_{j=1}^{n} \int_0^1 |A_{ij}(s)|^2 ds.
$$

An example of a family $A_{\sigma_{\min}}(s)$ is:

$$
A_{\sigma_{\min}}(s) = (1 - \sigma_{\min})A(s) + \sigma_{\min} I,
$$

where $A(s)$ satisfies the conditions in Assumption 4.1. We define the flow $\psi(\cdot|\epsilon)$ as

$$
\psi_s(\cdot|\epsilon) = A_{\sigma_{\min}}(s)\mathbf{x}_0 + (1 - A_{\sigma_{\min}}(s))\epsilon, \epsilon \sim \mathcal{N}(0, I).
$$

It's derivative is

$$
\psi'_s(\mathbf{x}|\epsilon) = A'_{\sigma_{\min}}(s)[\mathbf{x} - \epsilon]
$$

The inverse flow is

$$
\psi_s^{-1}(\mathbf{x}_s|\epsilon) = A_{\sigma_{\min}}(s)^{-1}(\mathbf{x}_s - \epsilon) + \epsilon,
$$

since $A_{\sigma_{\min}}(s)$ is invertible for $\sigma_{\min} > 0$. The marginal vector field $u(\mathbf{x}_s|\epsilon)$ is now defined as

$$
\begin{aligned}
u_s(\cdot|\epsilon) &= A'_{\sigma_{\min}}(s)A_{\sigma_{\min}}(s)^{-1}[\mathbf{x}_s - \epsilon] &(39) \\
&\equiv A'_{\sigma_{\min}}(s)[\mathbf{x}_0 - \epsilon], &(40)
\end{aligned}
$$

where we trivially use the result $A(s)^{-1}A(s) = I$. This vector field holds for all $\sigma_{\min} \in (0, 1)$, which is the domain in which $A_{\sigma_{\min}}(s)^{-1}$ is defined.

We note that $H^1([0, 1]; \mathbb{R}^{n \times n})$ is a Hilbert space, meaning it is *complete* with respect to the $\| \cdot \|_{H^1([0,1];\mathbb{R}^{n \times n})}$ norm. Because $A_{\sigma_{\min}}(s)$ is continuous in $\sigma_{\min}$ in this norm, the family of functions $\{A_{\sigma_{\min}}(s)\}_{\sigma_{\min} \in (0,1)}$ is *Cauchy* in $H^1([0, 1]; \mathbb{R}^{n \times n})$. As a result, there exists a limit function $\lim_{\sigma_{\min} \to 0} A_{\sigma_{\min}}(s) = A_0(s) \triangleq A(s)$ with convergence in the $\| \cdot \|_{H^1([0,1];\mathbb{R}^{n \times n})}$ norm.

By extension, we conclude that $\lim_{\sigma_{\min} \to 0} A'_{\sigma_{\min}}(s) = A'(s)$. Thus, the marginal vector field at $\sigma_{\min} = 0$ is

$$
u_s(\cdot|\epsilon) = A'(s)[\mathbf{x}_0 - \epsilon].
$$

## C   Summary of Baseline Models

We demonstrate strong performance against a wide range of TPP benchmarks. The authors of (Xue et al., 2024) provide implementations and evaluations for widely cited models in the field:

1. Recurrent marked temporal point process (RMTPP) (Du et al., 2016)

2. Neural Hawkes Process (NHP) (Mei & Eisner, 2017)

3. Self-attentive Hawkes process (SAHP) (Zhang et al., 2020)

4. Transformer Hawkes process (THP) (Zuo et al., 2020)

5. Attentive neural Hawkes process (AttNHP) (Mei et al., 2021)

6. Intensity-free TPP (IFTPP) (Shchur et al., 2020)

Another baseline method we include is Decomposable Transformer Point Processes (DTPP) (Panos, 2024). To ensure fairness in our comparisons, we modify its implementation by training the model to predict the inter-event time directly, rather than its logarithm, aligning it with the prediction approach used by other methods.

As the first paper to use diffusion models for TPPs, we included Add and Thin (Lüdke et al., 2023) to our baselines. Since Add and Thin only predicts the inter-event time instead of event type, we only reported the RMSE in Table 1. Consequently, we did not compute the OTD since event types are missing.

Since HYPRO (Xue et al., 2022) is an important baseline for long horizon prediction in the TPP literature, we also included it in our baselines.

We have showcased the performance of our proposed method by comparing it against these benchmarks.

## D    Reproducibility Details

We train the DiT model using a single NVIDIA P100 GPU. Despite the simplicity of the hardware setup, training remains computationally feasible: the full training process takes approximately one day to complete. We implement our model in PyTorch and use the Adam optimizer with a fixed learning rate of $1 \times 10^{-4}$. The model is trained with a batch size of 4, and all models are evaluated using the same random seed for reproducibility.

The DiT is configured with 7 layers (depth), 16 attention heads, and a hidden size of 1152. The latent representations output by the VAE encoder and decoded by the DiT have a dimensionality of 32. Each model is trained for 1,000,000 iterations, which ensures convergence for both short and long horizon prediction tasks across datasets of varying complexity. We define a diffusion temporal range $T_r = 10,000$ and a temporal context length $T_c = 128$. We define the argument $a \in \mathbb{R}^{N \times T_c}$ elementwise as $[a]_{ij} = [A(s)]_{ii} T_r^{-\frac{j-1}{T_c}}$ and construct the timestep embedding $e \in \mathbb{R}^{N \times (2T_c)}$ by $e = [\cos a, \sin a]$.

# E   VAE Training Results

We provide an ablation study on our choice for latent dimensions and regularization strengths $\beta_{max}$ by assessing the performance of a trained VAE on the TPP datasets. We evaluate the inter-event time reconstruction of the VAE using the mean squared error (MSE) metric, and the event type reconstruction using the accuracy metric.

We provide a separate table for the ablation study on the Retweet dataset. This is because the inter-event times take significantly larger values than the other datasets, which is why the MSE also takes larger values.

For our experiments, we decided on latent dimensions of size 16 and 32 and $\beta_{max}$ of 0.01 and 0.001, as they gave the best results in this study.

| Dataset | $d_{latent}$ | $\beta_{max}$ | Test MSE (Time) | Test Accuracy (Mark) | KL Divergence |
|---|---|---|---|---|---|
| Taxi | 8 | 0.01 | 0.000008 | 1.00 | 1.422255 |
| Taobao | 8 | 0.01 | 0.000007 | 1.00 | 2.675954 |
| Stackoverflow | 8 | 0.01 | 0.000007 | 1.00 | 1.302106 |
| Amazon | 8 | 0.01 | 0.000004 | 1.00 | 0.546956 |
| Taxi | 8 | 0.001 | 0.000066 | 1.00 | 1.595017 |
| Taobao | 8 | 0.001 | 0.000307 | 1.00 | 1.727479 |
| Stackoverflow | 8 | 0.001 | 0.000006 | 1.00 | 1.618665 |
| Amazon | 8 | 0.001 | 0.000003 | 1.00 | 1.749470 |
| Taxi | 16 | 0.01 | 0.000004 | 1.00 | 0.660862 |
| Taobao | 16 | 0.01 | 0.000001 | 1.00 | 0.510525 |
| Stackoverflow | 16 | 0.01 | 0.000006 | 1.00 | 0.972457 |
| Amazon | 16 | 0.01 | 0.000001 | 1.00 | 0.431824 |
| Taxi | 16 | 0.001 | 0.000002 | 1.00 | 1.175252 |
| Taobao | 16 | 0.001 | 0.000002 | 1.00 | 1.826852 |
| Stackoverflow | 16 | 0.001 | 0.000003 | 1.00 | 2.593421 |
| Amazon | 16 | 0.001 | 0.000001 | 1.00 | 0.486753 |
| Taxi | 32 | 0.01 | 0.000005 | 1.00 | 0.847024 |
| Taobao | 32 | 0.01 | 0.000001 | 1.00 | 1.100021 |
| Stackoverflow | 32 | 0.01 | 0.000006 | 1.00 | 0.969485 |
| Amazon | 32 | 0.01 | 0.000001 | 1.00 | 0.226177 |
| Taxi | 32 | 0.001 | 0.000002 | 1.00 | 2.596737 |
| Taobao | 32 | 0.001 | 0.000009 | 1.00 | 3.441910 |
| Stackoverflow | 32 | 0.001 | 0.000003 | 1.00 | 2.321191 |
| Amazon | 32 | 0.001 | 0.000004 | 1.00 | 0.286039 |
| Taxi | 8 | 0.1 | 0.000036 | 1.00 | 0.464769 |
| Taobao | 8 | 0.1 | 0.000040 | 1.00 | 0.367910 |
| Stackoverflow | 8 | 0.1 | 0.000029 | 1.00 | 0.639186 |
| Amazon | 8 | 0.1 | 0.000019 | 1.00 | 0.348057 |
| Taxi | 4 | 0.01 | 0.000114 | 0.999865 | 1.591100 |
| Taobao | 4 | 0.01 | 0.000279 | 1.00 | 1.710422 |
| Stackoverflow | 4 | 0.01 | 0.000169 | 1.00 | 2.114793 |
| Amazon | 4 | 0.01 | 0.000013 | 0.999988 | 1.121674 |
| Taxi | 8 | 1.0 | 0.000106 | 1.00 | 0.282435 |
| Taobao | 8 | 1.0 | 0.000076 | 1.00 | 0.337149 |
| Stackoverflow | 8 | 1.0 | 0.000363 | 1.00 | 0.645850 |
| Amazon | 8 | 1.0 | 0.000102 | 1.00 | 0.270750 |

Table 3: Overview of the training results of VAEs in four TPP benchmark datasets with different hyperparameters. $\beta_{max}$ corresponds to the weight we placed on the KL Divergence term.

| Dataset | $d_{latent}$ | $\beta_{max}$ | Test MSE (Time) | Test Accuracy (Mark) | KL Divergence |
|---------|--------------|---------------|-----------------|----------------------|---------------|
| Retweet | 2 | 1.0 | 0.002049 | 1.00 | 3.173542 |
| Retweet | 2 | 0.1 | 0.014252 | 0.999804 | 3.082116 |
| Retweet | 2 | 0.01 | 0.134966 | 0.591000 | 55.257951 |
| Retweet | 2 | 0.001 | 0.004303 | 0.542401 | 158.649788 |
| Retweet | 4 | 1.0 | 0.029607 | 0.999640 | 1.486775 |
| Retweet | 4 | 0.1 | NaN | 0.519590 | NaN |
| Retweet | 4 | 0.01 | 0.000839 | 0.999967 | 8.499786 |
| Retweet | 4 | 0.001 | NaN | 0.488553 | NaN |
| Retweet | 8 | 1.0 | 0.001032 | 1.00 | 0.667470 |
| Retweet | 8 | 0.1 | 0.004570 | 1.00 | 1.273117 |
| Retweet | 8 | 0.01 | 0.000254 | 1.00 | 1.857849 |
| Retweet | 8 | 0.001 | 0.000138 | 1.00 | 4.784689 |
| Retweet | 16 | 1.0 | 0.000290 | 1.00 | 0.276919 |
| Retweet | 16 | 0.1 | 0.000208 | 1.00 | 0.810816 |
| Retweet | 16 | 0.01 | 0.000250 | 1.00 | 1.547005 |
| Retweet | 16 | 0.001 | 0.000031 | 1.00 | 2.757766 |
| Retweet | 32 | 1.0 | 0.000648 | 1.00 | 0.185612 |
| Retweet | 32 | 0.1 | 0.000036 | 1.00 | 0.560295 |
| Retweet | 32 | 0.01 | 0.000081 | 1.00 | 1.235596 |
| Retweet | 32 | 0.001 | 0.000015 | 1.00 | 2.917406 |

Table 4: Training results of VAEs on the Retweet dataset. This table is posted separately because the time between events takes a larger range of values.

# F  Algorithms for Testing

In this section, we provide the method we used for forecasting future events conditioned on preceding events in an algorithmic format. We refer the reader to section 3.4 for a discussion that motivates our algorithms.

## F.1  Next Event Prediction Evaluation

We present our methods for next event prediction. This is obtained by solving the ODE defined element-wise in Equation 8. The observation window is $O = \{1, ..., n-1\}$ and the prediction window is $P = \{n\}$. This ensures that preceding events are reconstructed exactly and the model generates the $n$-th event. This is repeated for $n = 2, ..., N$.

The ODE is solved from $s = s_{end}^{(n)}$ to $s = s_{start}^{(n)}$, where $s_{end}^{(n)}$ and $s_{start}^{(n)}$ are defined in Equation 6. The initial condition is $\mathbf{x}(s_{end}^{(n)}) = A(s_{end}^{(n)})\mathbf{x}_0 + (I - A(s_{end}^{(n)}))\boldsymbol{\epsilon}$, where $\boldsymbol{\epsilon}$ is Gaussian noise. We provide the detailed method in Algorithm 1.

Additionally, we want the prediction of the $n$-th event to depend only on events $1, ..., n-1$. For this reason, we mask out events $n+1, ..., N$ so that the importance of proceeding events on the prediction of the $n$-th event is minimized.

---

**Algorithm 1** Next Event Prediction Evaluation

---

**Require:** Asynchronous noise schedule $A(s)$, pre-trained diffusion model $v_\theta(\mathbf{x}_s, A(s))$, pre-trained VAE $(E_\phi(\cdot), D_\psi(\cdot))$, test event sequence $\{\mathbf{z}^{(1)}, ..., \mathbf{z}^{(N)}\}$.
    Get latent event sequence $\mathbf{y} = \{\mathbf{y}^{(1)}, ..., \mathbf{y}^{(N)}\}$ where $\mathbf{y}^{(i)} = E_\phi(\mathbf{z}^{(i)})$ for $i = 1, ..., N$.
    **for** n=2,...,N **do**
        Sample noise $\boldsymbol{\epsilon} = \{\boldsymbol{\epsilon}^{(1)}, ..., \boldsymbol{\epsilon}^{(N)}\}$ where the dimension of $\boldsymbol{\epsilon}^{(i)}$ matches the dimension of $\mathbf{y}^{(i)}$ for all $i$.
        Initiate `mask` (shape: $[1, 1, N, N]$) where `mask[:,:,:n,:]=1` and `mask` is zero elsewhere.
        Define a vector field $f(\mathbf{x}_s, s)$ elementwise:

$$f_i(\mathbf{x}_s, s) = \begin{cases} [A'(s)]_{ii}(\mathbf{y}^{(i)} - \boldsymbol{\epsilon}^{(i)}) & i < n \text{ (Ensuring preceding events converge to } \mathbf{y}^{(i)}) \\ [A'(s)]_{ii}[v_\theta(\mathbf{x}_s, A(s), \texttt{mask})]_i & i = n \text{ (Predicting event } \mathbf{y}^{(n)}) \end{cases} \tag{41}$$

        Solve the ODE $\dot{\mathbf{x}}_s = f(\mathbf{x}_s, s)$ from $s = s_{end}^{(n)}$ to $s = s_{start}^{(n)}$ using an ODE solver with the initial condition $\mathbf{x}_{s_{end}^{(n)}} = A(s_{end}^{(n)})\mathbf{y} + (I - A(s_{end}^{(n)}))\boldsymbol{\epsilon}$.
        Obtain sequence $\{\mathbf{y}^{(1)}, ..., \mathbf{y}^{(n-1)}, \tilde{\mathbf{y}}^{(n)}\}$ consisting of preceding latent events $\{\mathbf{y}^{(1)}, ..., \mathbf{y}^{(n-1)}\}$ and predicted latent event $\tilde{\mathbf{y}}^{(n)}$.
        Decode $\tilde{\mathbf{y}}^{(n)}$: $D_\psi(\tilde{\mathbf{y}}^{(n)}) = \tilde{\mathbf{z}}^{(n)} = (\tilde{\tau}^{(n)}, \tilde{k}^{(n)})$.
    **end for**
    **Return** $RMSE(\{\tilde{\tau}^{(i)}, \tau^{(i)}\}_{i=2}^N), ErrorRate(\{\tilde{k}^{(i)}, k^{(i)}\}_{i=2}^N)$

---

## F.2  Optimal Transport Distance

We use the optimal transport distance (OTD) to assess the long horizon evaluation ADiff4TPP. We use the `distance_between_event_seq(.)` function posted in the GitHub repository of Mei et al. (2019). We use the following hyperparameters: `del_cost=1,trans_cost=1`. To generate events in a prediction horizon $h$, we solve the ODE defined element-wise in Equation 8. The observation window is $O = \{1, ..., N-h\}$ and the prediction window is $P = \{N-h+1, ..., N\}$. Our detailed method is posted in Algorithm 2.

---

**Algorithm 2** Long Horizon Prediction Evaluation

---

**Require:** Asynchronous noise schedule $A(s)$, pre-trained diffusion model $v_\theta(\mathbf{x}_s, A(s))$, pre-trained VAE $(E_\phi(\cdot), D_\psi(\cdot))$, test event sequence $\{\mathbf{z}^{(1)}, ..., \mathbf{z}^{(N)}\}$, prediction horizon $h$.

Get latent event sequence $\mathbf{y} = \{\mathbf{y}^{(1)}, ..., \mathbf{y}^{(N)}\}$ where $\mathbf{y}^{(i)} = E_\phi(\mathbf{z}^{(i)})$ for $i = 1, ..., N$.

Sample noise $\boldsymbol{\epsilon} = \{\boldsymbol{\epsilon}^{(1)}, ..., \boldsymbol{\epsilon}^{(N)}\}$ where the dimension of $\boldsymbol{\epsilon}^{(i)}$ matches the dimension of $\mathbf{y}^{(i)}$ for all $i$.

Define a vector field $f(\mathbf{x}_s, s)$ elementwise:

$$f_i(\mathbf{x}_s, s) = \begin{cases} [A'(s)]_{ii}(\mathbf{y}^{(i)} - \boldsymbol{\epsilon}^{(i)}) & i \leq N - h \text{ (Ensuring preceding events converge to } \mathbf{y}^{(i)}) \\ [A'(s)]_{ii}[v_\theta(\mathbf{x}_s, A(s))]_i & i > N - h \text{ (Predicting events } \tilde{\mathbf{y}}^{(N-h+1)}, ..., \tilde{\mathbf{y}}^{(N)}) \end{cases} \tag{42}$$

Solve the ODE $\dot{\mathbf{x}}_s = f(\mathbf{x}_s, s), \mathbf{x}_1 = \boldsymbol{\epsilon}$ from $s = 1$ to $s = 0$ using an ODE solver.

Obtain sequence $\{\mathbf{y}^{(1)}, ..., \mathbf{y}^{(N-h)}, \tilde{\mathbf{y}}^{(N-h+1)}, ..., \tilde{\mathbf{y}}^{(N)}\}$ consisting of preceding latent events $\{\mathbf{y}^{(1)}, ..., \mathbf{y}^{(N-h)}\}$ and predicted latent events $\{\tilde{\mathbf{y}}^{(N-h+1)}, ..., \tilde{\mathbf{y}}^{(N)}\}$.

Decode $\tilde{\mathbf{y}}^{(i)}$: $D_\psi(\tilde{\mathbf{y}}^{(i)}) = \tilde{\mathbf{z}}^{(i)} = (\tilde{\tau}^{(i)}, \tilde{k}^{(i)})$, for $i = N - h + 1, ..., N$.

**Return** `distance_between_event_seq`$(\{\tilde{\mathbf{z}}^{(i)}\}_{i=N-h+1}^N, \{\mathbf{z}^{(i)}\}_{i=N-h+1}^N)$

---

## G   Long Horizon Prediction Results

In this section, we compare the long horizon prediction of ADiff4TPP with existing baselines. We report the results of ADiff4TPP with 32 latent dimensions and $\beta_{\max} = 0.01$. We set the horizon length to 5, 10, 20, and 30, and report the mean and standard deviation of the OTD over all the datasets and models in five seeds. The results show that ADiff4TPP significantly outperforms all the other models. This is because ADiff4TPP initiates the generation of events in the more distant future before it completely generates events in the near future, thus sequentially providing stronger conditioning for future events.

Table 5: OTD (5 Events / 10 Events). Standard deviations are posted below. **Bold** indicates state-of-the-art results.

| Model | Amazon | Retweet | Taxi | Taobao | StackOverflow |
|---|---|---|---|---|---|
| RMTPP | 9.880/19.671 (0.016/0.035) | 9.991/19.983 (0.003/0.004) | 4.012/5.961 (0.136/0.196) | 8.692/15.808 (0.097/0.237) | 9.463/18.735 (0.075/0.160) |
| NHP | 9.881/19.670 (0.015/0.035) | 9.992/19.954 (0.003/0.008) | 4.007/5.915 (0.137/0.199) | 8.498/15.202 (0.099/0.248) | 9.482/18.762 (0.073/0.156) |
| SAHP | 9.874/19.657 (0.016/0.035) | 9.998/19.977 (0.002/0.006) | 5.458/7.566 (0.113/0.162) | 8.615/15.546 (0.101/0.247) | 9.490/18.790 (0.072/0.156) |
| THP | 9.869/19.642 (0.017/0.037) | 9.991/19.937 (0.003/0.010) | 6.023/10.816 (0.094/0.163) | 8.698/15.826 (0.095/0.234) | 9.498/18.808 (0.069/0.148) |
| AttNHP | 9.862/19.624 (0.018/0.039) | 9.981/19.925 (0.005/0.012) | 3.970/5.794 (0.127/0.166) | 8.216/14.659 (0.105/0.247) | 9.443/18.682 (0.073/0.157) |
| IFTPP | 9.859/19.603 (0.018/0.042) | 9.989/19.927 (0.003/0.011) | 4.461/5.573 (0.098/0.162) | 8.030/14.212 (0.104/0.245) | 9.406/18.626 (0.077/0.164) |
| DTPP | 6.848/13.734 (0.010/0.024) | 9.916/19.433 (0.004/0.009) | 2.983/6.773 (0.013/0.019) | 6.844/15.127 (0.019/0.032) | 7.554/14.946 (0.020/0.036) |
| HYPRO | 6.976/12.988 (0.016/0.055) | 9.491/19.653 (0.003/0.005) | 3.403/5.781 (0.023/0.034) | 5.786/11.367 (0.028/0.047) | 7.330/12.246 (0.017/0.028) |
| ADiff4TPP | **6.219/12.419** (0.049/0.115) | **9.119/17.738** (0.017/0.051) | **2.332/3.979** (0.022/0.015) | **5.398/10.255** (0.086/0.065) | **6.452/11.972** (0.400/0.922) |

Table 6: OTD (20 Events). Standard deviations are posted below. **Bold** indicates state-of-the-art results.

| Model | Amazon | Retweet | Taxi | Taobao | StackOverflow |
|---|---|---|---|---|---|
| RMTPP | 39.156 (0.075) | 39.906 (0.012) | 9.203 (0.326) | 28.727 (0.506) | 37.046 (0.320) |
| NHP | 39.161 (0.075) | 39.745 (0.028) | 9.170 (0.327) | 28.056 (0.530) | 37.188 (0.306) |
| SAHP | 39.065 (0.080) | 39.874 (0.020) | 12.784 (0.231) | 28.262 (0.523) | 37.232 (0.304) |
| THP | 39.069 (0.082) | 39.691 (0.032) | 14.308 (0.318) | 28.774 (0.499) | 37.116 (0.306) |
| AttNHP | 39.051 (0.083) | 39.694 (0.032) | 9.029 (0.240) | 27.046 (0.499) | 36.847 (0.323) |
| IFTPP | 38.971 (0.091) | 39.705 (0.031) | 8.566 (0.221) | 26.243 (0.503) | 36.763 (0.331) |
| DTPP | 27.603 (0.064) | 29.762 (0.003) | 13.830 (0.145) | 32.275 (0.136) | 29.188 (0.036) |
| HYPRO | 26.103 (0.140) | 30.749 (0.063) | 9.955 (0.056) | 20.805 (0.074) | 29.224 (0.454) |
| ADiff4TPP | **24.645** (0.141) | **28.028** (0.170) | **6.672** (0.049) | **19.152** (0.149) | **22.334** (2.142) |

Table 7: OTD (30 Events). Standard deviations are posted below. **Bold** indicates state-of-the-art results.

| Model | Amazon | Retweet | Taxi | Taobao | StackOverflow |
|---|---|---|---|---|---|
| RMTPP | 58.378 (0.120) | 59.792 (0.021) | 12.178 (0.449) | 40.602 (0.746) | 54.829 (0.487) |
| NHP | 58.390 (0.119) | 59.315 (0.056) | 12.108 (0.452) | 39.336 (0.766) | 55.191 (0.462) |
| SAHP | 58.270 (0.124) | 59.618 (0.038) | 17.474 (0.413) | 39.928 (0.770) | 55.226 (0.459) |
| THP | 58.233 (0.130) | 59.203 (0.063) | 17.553 (0.388) | 40.696 (0.735) | 54.905 (0.469) |
| AttNHP | 58.193 (0.130) | 59.171 (0.065) | 11.719 (0.307) | 38.404 (0.723) | 54.420 (0.492) |
| IFTPP | 58.119 (0.140) | 59.200 (0.064) | 11.592 (0.337) | 37.587 (0.703) | 54.311 (0.500) |
| DTPP | 41.745 (0.161) | 39.602 (0.063) | 20.669 (0.482) | 50.039 (0.684) | 43.399 (0.486) |
| HYPRO | 34.134 (0.155) | 38.952 (0.037) | 13.914 (0.072) | 30.686 (0.198) | 36.769 (0.563) |
| ADiff4TPP | **32.866** (0.477) | **31.653** (0.208) | **8.903** (0.012) | **29.181** (1.131) | **30.660** (1.789) |

# H   Ablation Studies on Diffusion Processes

## H.1   Different Noise Schedules

It is naturally important to assess whether diffusion models with asynchronous noise schedules outperform diffusion models without asynchronous noise schedules, and furthermore, whether there are noise schedules that outperform our choice of $A(s)$ in Equation 6. To evaluate this, we compare ADiff4TPP with diffusion models trained on two different noise schedules:

1. A **disjoint** noise schedule, where one event is diffused at a time. This is identical to autoregressive modelling of event sequences. The noise schedule is given as:

$$[A(s)^{disjoint}]_{ii} = \text{clip}\left(\frac{s_{end}^{(i,disjoint)} - s}{s_{end}^{(i,disjoint)} - s_{start}^{(i,disjoint)}}, \min = 0, \max = 1\right),$$

   where

$$s_{start}^{(i,disjoint)} = \frac{N-i}{N}, \quad s_{end}^{(i,disjoint)} = \frac{N-i+1}{N}.$$

2. A **synchronous** noise schedule, where all events are diffused at the same time. This is identical to rectified flow. The noise schedule is given as:

$$A(s)^{sync} = (1-s)I.$$

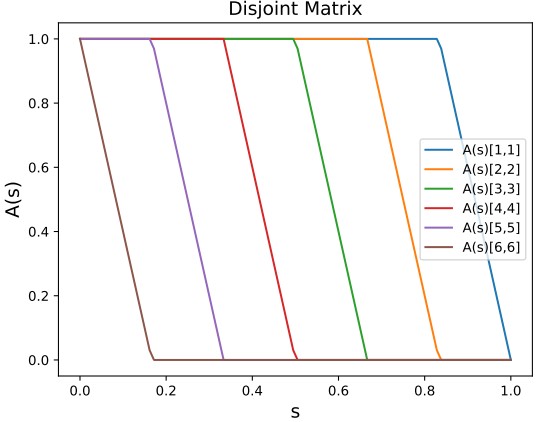

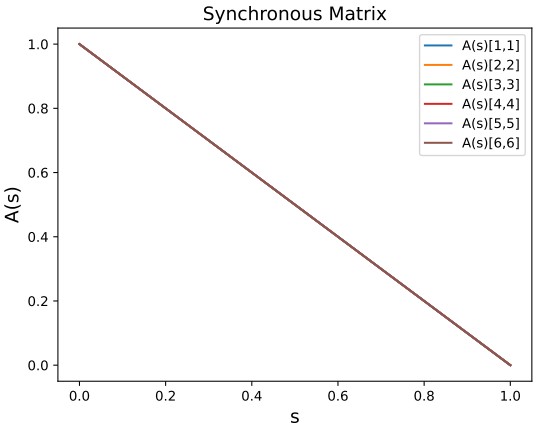

(a) An example of **disjoint** noise schedule with 6 events. The noise schedule shows that the event sequence diffuses one event at a time. Event $i$ starts diffusing after event $i+1$ is completely diffused.

(b) An example of **synchronous** noise schedule with 6 events. The noise schedule shows that every event starts diffusing at $s = 0$ and ends at $s = 1$.

Figure 5: Different noise schedules

Examples of $A(s)^{disjoint}$ and $A(s)^{sync}$ with 6 events are plotted in Figure 5. We train diffusion models with 32 latent dimensions and $\beta_{\max} = 0.01, 0.001$ on the five TPP datasets for both of these noise schedules and compare them with ADiff4TPP in next event prediction tasks.

### H.1.1   Number of Function Evaluations

We also compare the computational efficiency of different noise schedules by reporting the number of function evaluations (NFEs) required during generation. NFEs correspond to the number of evaluations of the DiT

during numerical integration of the reverse-time ODE along a prediction horizon $h$. We use RK4 as the solver, which requires 4 NFEs per integration step.

The effective number of NFEs depends on the interval of the flow time over which the ODE must be solved. In synchronized diffusion, the reverse ODE is solved over the full interval $[0, 1]$ for all event indices. In ADiff4TPP, each event is generated over a shorter sub-interval $[s_{start}^{(i+h)}, s_{end}^{(i)}]$, reducing the effective integration range. In disjoint diffusion, the ODE for each event is solved only over the small interval $[(N-(i+h)-1)/N, (N-i)/N]$, resulting in the shortest integration range.

For prediction along a horizon of length $h$ using RK4, the effective ODE ranges imply:

- Synchronized diffusion: $4(2N-1)$ NFEs

- ADiff4TPP: $4N + 4(h-1)$ NFEs

- Disjoint diffusion: $4h$ NFEs

Importantly, synchronized diffusion requires solving over the full diffusion interval regardless of the prediction horizon, while autoregressive-style methods scale linearly with the horizon length $h$.

## H.2 Absence of Masking

We assessed the importance of masking by comparing the results of ADiff4TPP in next event prediction tasks with and without masking.

The removal of masking is trivally done by modifying Algorithm 1 so that the vector field $f(\mathbf{x}(s), s)$ from equation 41 is instead defined elementwise as:

$$f_i(\mathbf{x}(s), s) = \begin{cases} A'(s)(\epsilon_i - \mathbf{x}_i) & i < n \\ A'(s)v_\theta(\mathbf{x}(s), A(s)) & i \geq n, \end{cases} \tag{43}$$

thus removing the masking in the computation of $v_\theta(\mathbf{x}(s), A(s))$. After solving the ODE, we then obtain a sequence $\{\mathbf{x}_1, ..., \mathbf{x}_{n-1}, \tilde{\mathbf{x}}_n, ..., \tilde{\mathbf{x}}_N\}$ consisting of preceding latent events $\{\mathbf{x}_1, ..., \mathbf{x}_{n-1}\}$ and predicted latent events $\{\tilde{\mathbf{x}}_n, ..., \tilde{\mathbf{x}}_N\}$ along the entire prediction horizon. We only decode $\tilde{\mathbf{x}}_n$.

The main difference is that the prediction of $\tilde{\mathbf{x}}_n$ is dependent only on past events if we use masking. If we remove masking, then the prediction of $\tilde{\mathbf{x}}_n$ depends on observations of past and future events.

## H.3 Results

We posted our ablation results in Table 2. ADiff4TPP with 32 latent dimensions outperforms all the other methods in next event predictions in the Amazon, Retweet, and Taobao datasets. ADiff4TPP with 16 latent dimensions and $\beta_{\max} = 0.01$ achieves the best results in the Stackoverflow dataset.

While Unmasked Diffusion shows comparable results to ADiff4TPP on a few datasets, it fails to match the consistency of ADiff4TPP across all metrics and datasets. In contrast, Synchronized Diffusion and Disjoint Diffusion perform significantly worse, with higher RMSEs and error rates. These results showcase the reliability of masking and asynchronous noise schedules in ADiff4TPP for prediction tasks in temporal point processes.

# I Ablation Studies on Event Representations

To evaluate whether the VAE is necessary, we compare the proposed $\beta$-VAE representation with two alternative event encoding strategies:

(i) a deterministic autoencoder (AE) with the same architecture but without KL regularization, and

(ii) a standard embedding consisting of a learnable mark embedding and a Time2Vec (Kazemi et al., 2019) encoding for inter-event time.

All models use a latent dimension of 32. We report the reconstruction results and the empirical KL divergence, which is used to measure the deviation of the latent vectors from Gaussian random vectors. This is computed as

$$\frac{1}{2} \sum_{i=1}^{d} (\sigma_i^2 + \mu_i^2 - 1 - \log(\sigma_i)),$$

where

$$\mu_i = \frac{1}{n} \sum_{j=1}^{n} z_j^i, \quad \sigma_i^2 = \frac{1}{n-1} \sum_{j=1}^{n} (z_j^i - \mu_i)^2,$$

and $z_j^i$ is the $j$-th component of the $i$-th latent vector. We report the event representation training results in Table 8.

Table 8: Event representation ablation. Reconstruction performance and deviation from a Gaussian prior.

| Model | Dataset | MSE (Time) | Accuracy (Mark) | KL Divergence |
|---|---|---|---|---|
| AE | Amazon | 0.000004 | 1.00 | 11.06 |
| | Retweet | 0.000577 | 1.00 | 35.62 |
| Std. Embedding | Amazon | 0.000011 | 1.00 | 371.83 |
| | Retweet | 0.038600 | 1.00 | 323.87 |

While both embedding strategies achieve strong reconstruction performance on Amazon and slightly worse performance on Retweet, we argue that reconstruction accuracy alone is insufficient for latent diffusion. The critical factor is the geometry and distributional structure of the latent space. To evaluate the practical impact, we trained DMs using the alternative embeddings and measured long-horizon prediction performance and post our results in Table 9.

Table 9: Long-horizon prediction performance using alternative event representations. Lower is better.

| Model | Dataset | 5 Events | 10 Events | 20 Events | 30 Events |
|---|---|---|---|---|---|
| AE | Amazon | 6.938 | 13.080 | 27.666 | 33.303 |
| | Retweet | 9.593 | 18.592 | 30.441 | 36.732 |
| Std. Embedding | Amazon | 6.732 | 13.664 | 29.009 | 34.225 |
| | Retweet | 9.623 | 19.190 | 31.078 | 36.972 |

The results show that DMs under these alternate embeddings show worse long horizon performance compared to ADiff4TPP and sometimes worse than HYPRO/DTPP. The KL divergence values provide insight into this behaviour. The deterministic AE already exhibits noticeable deviation from a Gaussian prior, while the standard embedding produces substantially larger deviations. In contrast, the VAE explicitly regularizes the latent distribution toward a unit Gaussian prior, resulting in a smoother and more uniformly supported latent space.

This alignment is particularly important for diffusion-based generation. A Gaussian-aligned latent space ensures:

- Local decodability (Barquero et al., 2023): small perturbations remain within the decodable manifold.

- Stable score/flow estimation (Lee et al., 2025): reverse diffusion does not require mapping from highly irregular distributions.

- Efficient sampling (Lee et al., 2025): the forward noising process begins closer to the assumed prior distribution.

Overall, these ablations indicate that the benefit of the VAE is not merely improved reconstruction, but improved latent geometry. The variational regularization produces a smoother and better-aligned latent space, which is more suitable for diffusion-based generative modelling.

