# OpenReview forum: "ADiff4TPP: Asynchronous Diffusion Models for Temporal Point Processes"
_TMLR — Accepted by TMLR_

### Review · Reviewer_wNCA · 2026-02-17

**Summary Of Contributions:**

I would like to note that I am not an expert in the field and did not publish any work related to TPPs. However, I have some background in generative modeling.

**Contributions**

The authors highlight several limitations in applying diffusion models to the modeling of TPPs. This includes data heterogeneity, a dependence of future observations on past events, and the fixed context length of diffusion models.

They address the first problem through latent diffusion and use asynchronous noise schedules to model varying uncertainty levels with increasing time and address the fixed context length of LLMs.

They demonstrate the effectiveness of their framework on common benchmark datasets.

**Strengths**

- The problems identified of the authors all have practical relevance in applying diffusion models to the generation of TPPs.
- I specifically liked removing the sequential nature of next even prediction at evaluation time
- The authors demonstrate that their introduced noise schedules are valid in the context of flow matching and preserve relevant properties
- As far as I could see from comparing to similar papers the authors use relevant benchmarks to validate their approach
- They ablate individual components that they introduce and show their effectiveness. However, some of the modifications only lead to minor improvements and its unclear how well some of these contributions generalize.


**Weaknesses**

- The abstract does not clearly describe the gap in current methods that the paper addresses. While contributions are stated, it remains unclear how the proposed method relates to previous work. The problem's importance is also not motivated.
- From the description in the introduction, it was unclear to me how the varying noise schedules address the fixed context problem.
- Figures 1 and 2 are highly redundant
- Wrapfigures could be consistently put to the right
- I am not well aware of previous literature in the field but I would strongly recommend the authors to not only provide their own baselines but actually compare to previous methods in the literature, including those not using diffusion models

**Audience:**

Yes

**Audience Explanation:**

The authors provide a clear description of the existing limitations in applying diffusion models to TPP generation and propose multiple methods to address them. I believe the findings in this paper are relevant to the community

**Broader Impact Concerns:**

Not applicable to this work.

**Claims And Evidence:**

Yes

**Claims Explanation:**

The authors provide evidence for the effectiveness of all the components they propose. They also mathematically demonstrate that their approach is sensible in the context of flow matching.

However, the paper lacks empirical comparisons to previous work.

**Requested Changes:**

- The abstract needs to motivate the importance of the problem that the authors aim to resolve
- The abstract needs to clearly state limitations in existing approaches. This includes limitations beyond applying Diffusion Models to TPPs but also why this is something we should be interested in
- The paper requires more empirical evidence and comparisons to previous methods (also those that do not rely on diffusion)
- My stated weaknesses regarding formatting should be addressed

---

> ### Author Response · Authors · 2026-02-19
>
> We sincerely thank the Reviewer for their encouraging comments and constructive suggestions. We address each point in turn below. Should there be any further questions, we will be happy to provide clarifications.
>
> **Abstract - Motivation for DMs to TPPs**
> We thank the reviewer for highlighting the need to better motivate the use of DMs for TPP modelling. The canonical task in TPPs is to learn the distribution of future events conditioned on past observations. Most existing approaches learn this distributions by relying on strong parametric assumptions (for example, conditional intensity functions in most benchmark methods, and mixture of Gaussians in IFTPP). This can limit expressivity and make long horizon forecasting expensive due to repeated next event prediction.
>
> Diffusion models, in contrast, are designed to learn complex data distributions without relying on parametric forms. In our setting, this allows modelling the *joint distribution* of event sequences in latent space, enabling long horizon forecasting within a single generative process. Theorem 4.5 formalizes that our model also learns the *conditional distribution* of future events provided past events, aligning with the objective in TPP modelling. We will revise the abstract to more clearly emphasize this modelling distinction, the practical implications for long horizon forecasting, and how our approach relates to prior TPP methods.
>
> **Flexible observation/prediction window**
> We appreciate the reviewer's request for clarification regarding how our method supports flexible observation and prediction windows. While the asynchronous noise schedule is important to this capability, it works in conjunction with our overall design (VAE + DiT + $A(s)$).
>
> We intentionally adopt a design that separates event representation (VAE) and the modelling of temporal dependencies (DiT). This separation allows us to operate directly in latent space and selectively control which event indices are treated as observed or predicted. During forecasting, latent variables corresponding to the observation window remain fixed, while those in the prediction window are initialized to Gaussian noise.
>
> The asynchronous noise schedule $A(s)$ enables this mechanism by assigning different start and end flow times to different event indices. By choosing an appropriate flow time $s$, we ensure that all events in the prediction window are fully diffused (i.e., Gaussian noise), while events in the observation window remain (partially) undiffused upon initialization. In reverse time, this yields conditional generation of future events given past events.
>
> This design allows arbitrary observation/prediction splits at inference time without retraining the model, enabling both short and long horizon forecasting with the same model and method. We will revise the abstract and the introduction with this explanation.
>
> **Figures**
> We thank the reviewer for the suggestion and will move Figure 3 to the right in the final version. We agree that this looks more professional and visually appealing.
>
> Regarding Figures 1 and 2, we acknowledge that they have redundancies. However, they serve complementary explanatory purposes: Figure 1 illustrates the proposed asynchronous schedule in detail, while Figure 2 provides a direct visual contrast with the synchronous diffusion setting. We believe that presenting both significantly improves accessibility, especially for readers less familiar with diffusion-based modelling. For this reason, we apologize to the reviewer, but we prefer to retain both figures in the final version.
>
> **Baseline Methods**
> We apologize if this was unclear in the main text. All methods in Table 1 are previously published baselines and not variants of our own model.
>
> We clarify that Table 1 compares ADiff4TPP against nine established baselines, including intensity-based (RMTPP, NHP), transformer-based (THP, SAHP), hybrid models (HYPRO), and diffusion-based (Add-Thin) methods. These implementations were reproduced, either by using the EasyTPP repository or by running publicly released code, to ensure fair comparison. We will revise Appendix C in the paper to more clearly highlight this point.

---

> > ### Comment · Reviewer_wNCA · 2026-02-19
> > **Thanks for the clarifications!**
> >
> > Dear authors,
> >
> > Thank you for the detailed response.
> >
> > **Abstract** Do you plan to incorporate some of this motivation in the abstract? I think 1-2 sentences are already sufficient to indicate to the reader, where the current gap lies and how you address it and why this is important.
> >
> > **Flexible prediction window** This gets clear in later stages of the paper. I still think the respective part in the introduction can be improved a bit.
> >
> > **Figures** I see the advantage and its not a crucial point.
> >
> > **Baseline Methods** I read the respective section again and actually think this was an error on my end and already clear enough in the original versions. Thanks for clarifying.
> >
> > I dont have any major remaining concerns.

---

> > > ### Author Response · Authors · 2026-02-22
> > > **Thank you for your helpful suggestions**
> > >
> > > Dear Reviewer,
> > >
> > > Thank you for your thoughtful follow-up and for carefully reconsidering our responses. We appreciate your constructive feedback and are glad that the clarifications addressed your main concerns.
> > >
> > > In the revised version, we will incorporate 2 additional sentences in the abstract to more clearly articulate (1) the modelling gap in existing TPP approaches (particularly the reliance on parametric assumptions and repeated next event prediction) and (2) how ADiff4TPP addresses this gap (joint multi-event and long-horizon forecasting).
> > >
> > > We will also improve the corresponding discussion in the introduction to more explicitly explain how the asynchronous noise schedule enables flexible observation and prediction window splits at inference time.
> > >
> > > Thank you again for your suggestions.
> > > The Authors

---

### Review · Reviewer_fwpP · 2026-02-22

**Summary Of Contributions:**

The paper proposes a diffusion-based generative framework for temporal point processes centered on an asynchronous (matrix-valued) noise schedule that injects different noise levels into different events depending on temporal order, with the goal that earlier future events “stabilize” sooner and provide stronger conditioning for later ones. The model operates in a continuous latent space learned by a $\beta$-VAE that encodes each event’s inter-event time and mark into a vector, and then uses a Transformer-style backbone (DiT) trained with an asynchronous flow-matching objective to model the joint distribution of latent event sequences.
Empirically, the paper reports strong next-event and long-horizon forecasting results, and includes ablations examining (i) latent dimension choices and (ii) schedule variants (asynchronous vs synchronous vs disjoint) plus the role of attention masking.

**Audience:**

Yes

**Audience Explanation:**

Diffusion in the latent space is a well-established area and has shown great promise. The work explores a nonstandard design choice and reports empirical gains plus ablations that may generalize to other sequence-generation settings.

**Claims And Evidence:**

Yes

**Claims Explanation:**

The paper reports that asynchronous schedules outperform synchronous and disjoint variants in Table 2, and interprets this as balancing joint modeling with reduced error propagation; it also shows masking helps. It also gives a concrete formulation for encoding mixed continuous/categorical events and argues that latent-space diffusion is convenient for mixed-type data.

**Requested Changes:**

1. While the ablation in this paper is reasonable, the most convincing version would keep the architecture, latent representation, and masking identical and swap only the schedule, while reporting both performance and wall-clock (or at least number of function solver steps) to substantiate any efficiency narrative.
2. In addition to the current latent-space ablations, it would really help to include at least one strong “no-VAE” alternative and run it through the exact same flow/diffusion backbone. That kind of baseline would make the “why do we need a VAE here?” story much more concrete and easy to judge.
3. It would help to slightly recalibrate the framing in the introduction and conclusion so the claims line up perfectly with what you actually show. In particular, I would spell out clearly what the model is generating in your experiments, and I would be explicit about what isn’t demonstrated yet, like truly arbitrary masked conditioning or explicitly modeling the event-count distribution.

---

> ### Author Response · Authors · 2026-02-24
> **Official Comment by Authors**
>
> We thank the reviewer for their generous and positive comments on the paper. We address each point below. We will be happy to provide clarifications if there are any further questions.
>
> **Ablation - Noise Schedule**
> We clarify that in Section 5.3 and Appendix H.1, the architecture, latent representation, and masking are kept identical across schedule variants; only the noise schedule is changed. The ablation includes:
> * Synchronized diffusion, where all events share the same noise level at all times (see Figure 5b or Figure 2).
> * Disjoint diffusion, where only one event is denoised at a time (see Figure 5a).
> We will add references to the figures in Section 5.3 to provide better intuition to the reader. And we will clarify that all other design choices are kept identical.
>
> Thank you for the NFE (number of function evaluations) suggestion. We will add that to section 5.3 as well. Synchronized diffusion requires more NFEs than ADiff4TPP because we are solving the ODE along $[0,1]$ instead of a sub-interval $[s_{start}^{(i)},s_{end}^{(i)}]$. Disjoint diffusion requires less NFEs because we are solving the ODE along a shorter $[(N-i-1)/N,(N-i)/N]$. Using RK4 (4 NFEs per step), the effective ODE ranges imply:
> * Synchronized diffusion: $4(2N-1)$ NFEs
> * ADiff4TPP: $4N$ NFEs
> * Disjoint diffusion: $4$ NFEs
>
> **Ablation - No VAE**
> We thank the reviewer for the suggestion. The role of the VAE in our framework is not merely architectural convenience but addresses a fundamental modelling issue: TPPs involve mixed continuous (inter-arrival time) and categorical (mark) variables, whereas DMs are naturally defined in continuous spaces.
>
> Removing the VAE would require either directly diffusing one-hot categorical variables as continuous vectors, or adopting discrete diffusion mechanisms. We see both cases as substantially different modelling frameworks, and believe the resulting model would no longer represent a controlled ablation that isolates the effect of the VAE while keeping the diffusion backbone unchanged.
>
> For this reason, we view the VAE as an integral component enabling continuous latent diffusion for mixed-type events. That said, we agree that exploring discrete or hybrid diffusion approaches for TPPs without latent encoding is an interesting direction for future work.
>
> **Introduction**
> We agree with the reviewer that the introduction can benefit from more explanation. A similar point was raised by Reviewer wNCA as well. We will revise it to more explicitly state that our model generates sequences of events in a latent space, with masked conditioning over a designated prediction window. The objective in TPP modelling is to learn the distribution of future events conditioned on past observations. While DMs are designed to learn the joint distribution of events in a sequence, we prove (Theorem 4.5) that they are also capable of learning the conditional distribution of future events provided past events.
>
> We will also clarify the current scope of the experiments in the conclusion. We will add that DiT with arbitrary masked conditioning (beyond prefix-based forecasting) and DMs for modelling the event-count distribution are areas for future work.

---

### Review · Reviewer_hMt6 · 2026-03-02

**Summary Of Contributions:**

1. This paper uses a beta-VAE to map both the continuous inter-event times and discrete categorical marks into a continuous latent space, resolving the challenge of modeling coexisting mixed data types. Within this latent space, it performs a diffusion model guided by an asynchronous time schedule and designs a specific conditional flow-matching training objective for this schedule. This asynchronous mechanism intuitively aligns with the sequential, causal nature of TPPs, where past events inherently influence future ones.

2. Extensive experiments empirically demonstrate the overall effectiveness of the proposed framework combining the VAE, flow matching, and asynchronous schedule. Furthermore, by explicitly comparing the proposed method against two extreme design alternatives—a fully synchronous schedule and a disjoint (autoregressive) schedule—the paper specifically validates the utility and necessity of the asynchronous schedule within this VAE+flow architecture.

**Audience:**

Yes

**Audience Explanation:**

TPP is a relevant topic for machine learning community.

**Claims And Evidence:**

No

**Claims Explanation:**

1. Justification for the VAE: The paper lacks justification for choosing a VAE over simpler, common encodings. Because VAEs introduce extra complexity and potential reconstruction errors (especially for continuous time), please provide an ablation study comparing the VAE against standard embeddings under a comparable computational budget to prove its necessity.

2. Clarification on efficiency: While efficiency is claimed multiple times in this paper, it currently lacks sufficient experimental backing. Please report the exact NFEs and wall-clock time the ODE solver requires to match the accuracy (e.g., OTD) of baselines, particularly synchronous diffusion models.

3. Isolating the async schedule's effectiveness: The asynchronous schedule is currently validated only within this specific VAE+flow setup. To prove its fundamental robustness, please evaluate it in a different framework (e.g., standard embeddings + flow) by comparing the async, sync, and discrete schedules.

4. In the event prediction task, the paper adopts RMSE as the evaluation metric. However, this metric is inherently problematic (although it is commonly used in many prior works). A temporal point process is stochastic — given the history, the model outputs a distribution over the next event. Therefore, the appropriate metric should measure the distance between the predicted distribution and the ground-truth distribution, rather than the distance between a predicted point and a ground-truth point. In fact, if the RMSE between two points approaches zero, this is itself questionable and does not necessarily indicate better performance.

**Requested Changes:**

See the 4 questions above.

---

> ### Author Response · Authors · 2026-03-03
> **Official Comment by Authors**
>
> We thank the reviewer for the detailed and constructive feedback. We appreciate the careful summary of our contributions and the recognition of the strengths of our approach. We address the remaining concerns below.
>
> **Justification for VAE**
> We thank the reviewer for this suggestion. To directly assess whether the variational formulation is necessary, we are conducting additional ablation studies comparing the VAE with two alternative event representation strategies under comparable computational settings, and will post the ablation results once obtained. Thank you for your patience.
>
> **NFE Comparison**
> We thank the reviewer for requesting a more concrete analysis of efficiency. We will include both the exact number of function evaluations (NFEs) and wall-clock runtime in the revised manuscript.
>
> For next event prediction, the NFEs for different schedules are:
> * Synchronized diffusion: $4(2N-1)$ NFEs
> * ADiff4TPP: $4N$ NFEs
> * Disjoint diffusion: $4$ NFEs
>
> where N denotes the sequence length and RK4 is used (4 NFEs per step).
>
> For long horizon prediction of length h:
> * Synchronized diffusion: $4(2N-1)$ NFEs
> * ADiff4TPP: $4N+4(h-1)$ NFEs
> * Disjoint diffusion: $4h$ NFEs
>
> Importantly, synchronized diffusion requires solving over the full interval regardless of horizon, while autoregressive methods scale linearly with h. We will report both NFEs and wall-clock time for $h=5,10,20,30$ to match this claim.
>
> **Async Schedule Effectiveness**
> We appreciate the reviewer's suggestion to evaluate the asynchronous schedule in alternative frameworks. We clarify that our empirical claim is limited to diffusion-based joint generation settings.
>
> All baselines in our experiments are autoregressive models that generate events sequentially by repeatedly predicting the next event and appending it to the sequence. In contrast, the asynchronous schedule is designed for joint multi-event generation within a single diffusion process. Incorporating it into purely autoregressive architectures would require redesigning their generation mechanism, rather than constituting a direct ablation.
>
> Within our framework, we isolate the effect of the schedule by keeping the architecture, latent representation, and masking fixed while varying only the schedule (synchronous, asynchronous, disjoint). Thus, the reported gains directly reflect the contribution of the asynchronous schedule in the intended joint-generation setting.
>
> We will revise the introduction and conclusion to clarify that our empirical claims regarding robustness are restricted to diffusion-based TPP modelling.
>
> **Evaluation metric**
> We thank the reviewer for the interesting suggestion. We agree that TPPs are inherently stochastic and that distribution-level metrics are ideal for evaluation.
>
> We adopt RMSE for next event prediction to remain consistent with established benchmarking protocols in neural TPP literature, ensuring direct comparability with prior work. We acknowledge that RMSE evaluates point estimates rather than full predictive distributions.
>
> To complement this, we evaluate long horizon forecasting using Optimal Transport Distance (OTD), which measures the distance between predicted and ground-truth event distributions (see [1]). In this setting, ADiff4TPP achieves state-of-the-art OTD results, suggesting that it is better at capturing the conditional distribution of future events given the past.
>
> We will clarify in the revised manuscript that RMSE is used for comparability, while OTD provides a distribution-level evaluation, and we will temper the language to avoid implying deterministic prediction.
>
> [1] Mei, Hongyuan et al. "Imputing missing events in continuous-time event streams". ICML 2019.

---

> > ### Author Response · Authors · 2026-03-04
> > **Justification for the VAE**
> >
> > We thank the reviewer for their patience. To assess whether the VAE formulation is necessary, we conducted additional ablations comparing the VAE with two alternative event representation strategies. In all cases, we keep the latent dimension, optimizer, and training budget fixed and only vary the event representation model.
> >
> > We report the reconstruction results and the empirical KL divergence, which is used to measure the deviation of the latent vectors from Gaussian random vectors. This is computed as $\frac{1}{2}\sum_{i=1}^{d}(\sigma_i^2+\mu_i^2-1-\log(\sigma_i))$, where $\mu_i=\frac{1}{n}\sum_{j=1}^{n}z_j^i$, $\sigma_i^2=\frac{1}{n-1}\sum_{j=1}^{n}(z_j^i-\mu_i)^2$, and $z_j^i$ is the $j$-th component of the $i$-th latent vector.
> >
> > (1) Deterministic Autoencoder (AE) vs. VAE
> >
> > We replace the VAE with a deterministic autoencoder of identical architecture, removing only the KL regularization term (equivalent to $\beta=0$). We report reconstruction performance and the empirical KL divergence of the learned latent distribution from a unit Gaussian.
> >
> > Reconstruction results:
> > |Dataset|Latent dimension|Test MSE (Time)|Test Accuracy (Mark)|KL Divergence|
> > |-|-|-|-|-|
> > |Amazon|32|0.000004|1.00|11.064751|
> > |Retweet|32|0.000577|1.00|35.617392|
> >
> > (2) Standard Embedding: Mark Embedding + Time2Vec
> >
> > We also evaluate a commonly used embedding strategy consisting of a learnable mark embedding and a Time2Vec encoding [1] for the inter-event time. The concatenated representation is projected to the same latent dimension and trained with a decoder for time and mark reconstruction.
> >
> > Reconstruction results:
> > |Dataset|Latent dimension|Test MSE (Time)|Test Accuracy (Mark)|KL Divergence|
> > |-|-|-|-|-|
> > |Amazon|32|0.000011|1.00|371.8342|
> > |Retweet|32|0.038600|1.00|323.8717|
> >
> > While both embedding strategies achieve strong reconstruction performance on Amazon and slightly worse performance on Retweet, we argue that reconstruction accuracy alone is insufficient for latent diffusion. The critical factor is the geometry and distributional structure of the latent space.
> >
> > Diffusion models inject Gaussian noise into latent codes and learn to reverse this perturbation process. If the latent space is sparse or strongly non-Gaussian, small perturbations may move latent points into regions poorly supported by the decoder, which can destabilize score/flow estimation and increase the burden on the reverse diffusion process [2,3].
> >
> > To evaluate the practical impact, we trained diffusion models using the alternative embeddings and measured long-horizon prediction performance.
> >
> > Using AE:
> > |Dataset|5 Events|10 Events|20 Events|30 Events|
> > |-|-|-|-|-|
> > |Amazon|6.938|13.080|27.666|33.303|
> > |Retweet|9.593|18.592|30.441|36.732|
> >
> > Using standard embedding:
> > |Dataset|5 Events|10 Events|20 Events|30 Events|
> > |-|-|-|-|-|
> > |Amazon|6.732|13.664|29.009|34.225|
> > |Retweet|9.623|19.190|31.078|36.972|
> >
> > The results show that DMs under these alternate embeddings show worse long horizon performance compared to ADiff4TPP and sometimes worse than HYPRO/DTPP.
> >
> > The KL divergence values provide insight into this behavior. The deterministic AE already exhibits noticeable deviation from a Gaussian prior, while the standard embedding produces substantially larger deviations. In contrast, the VAE explicitly regularizes the latent distribution toward a unit Gaussian prior, resulting in a smoother and more uniformly supported latent space.
> >
> > This alignment is particularly important for diffusion-based generation. A Gaussian-aligned latent space ensures:
> > * Local decodability [2]: small perturbations remain within the decodable manifold.
> > * Stable score/flow estimation [3]: reverse diffusion does not require mapping from highly irregular distributions.
> > * Efficient sampling [3]: the forward noising process begins closer to the assumed prior distribution.
> >
> > Overall, these ablations indicate that the benefit of the VAE is not merely improved reconstruction, but improved latent geometry. The variational regularization produces a smoother and better-aligned latent space, which is more suitable for diffusion-based generative modelling.
> >
> > We will include these ablation results and clarifications in the revised manuscript to more clearly justify the role of the VAE within our diffusion framework.
> >
> > [1] Kazemi, Seyed Mehran, et al. "Time2vec: Learning a vector representation of time." arXiv preprint arXiv:1907.05321 (2019).
> >
> > [2] Barquero, German, et al. "Belfusion: Latent diffusion for behavior-driven human motion prediction." Proceedings of the IEEE/CVF international conference on computer vision. 2023.
> >
> > [3] Lee, Junho, et al. "Latent diffusion models with masked autoencoders." Proceedings of the IEEE/CVF International Conference on Computer Vision. 2025.

---

### Author Response · Authors · 2026-03-16
**Updates based on Reviewer Feedback**

We thank the reviewers again for their constructive feedback. In response to the comments, we have revised the manuscript to improve clarity, strengthen the theoretical presentation, and expand the empirical evaluation. The main changes in the revised version (highlighted in blue) are summarized below.

**Improved clarity in the abstract and introduction**

We updated the abstract to provide a clearer motivation for applying DMs to TPPs, and to better highlight the limitations of existing approaches. We also revised parts of the introduction and the methodological description of asynchronous noise schedules to better motivate the design of ADiff4TPP.

**Additional analysis of noise schedules**

Appendix H has been expanded to provide a clearer comparison of different noise schedules. In particular, we added a new subsection comparing the number of function evaluations (NFEs) required for each schedule, highlighting the computational efficiency of the proposed asynchronous formulation.

**Ablation - Justification for the VAE**

We conducted additional ablation studies on long-horizon prediction using two alternative event representations: a deterministic autoencoder (i.e., $\beta = 0$) and a standard embedding-based representation (Time2Vec combined with a mark embedding layer). While both approaches achieve nearly perfect reconstruction accuracy, the results indicate that the VAE encourages a latent geometry better aligned with the Gaussian assumptions used in diffusion models. Consequently, the deterministic representations lead to weaker performance on long-horizon prediction.

Thank you once again for your careful consideration and helpful feedback. If any questions remain, or if any clarification would be useful, we would be very grateful to hear from you.

Sincerely,

---

### Decision · Action_Editor_zQk5 · 2026-04-07

**Recommendation:** Accept with minor revision

**Additional Comments:**

Reviewers recommend Accept, Accept, and Leaning Accept. The paper makes a coherent contribution, and the rebuttal adequately addressed major concerns. Minor revisions should include: (1) updating the abstract/introduction per agreed revisions; (2) incorporating VAE ablations and NFE analysis into the manuscript; (3) adjusting efficiency and generalization claims to match empirical scope.

**Audience:**

Yes

**Audience Explanation:**

The paper addresses diffusion/flow-matching for temporal point processes, both active ML areas.

**Claims And Evidence:**

Yes

**Claims Explanation:**

All three reviewers agreed the claims are supported. The authors provided theoretical justification (Theorem 4.5), empirical benchmarks, and responded to reviewer concerns with additional ablations (VAE vs. AE vs. standard embeddings), NFE comparisons, and clarified evaluation metrics. Evidence is sufficient and consistent with TMLR standards.